# Reduction of corpus callosum activity during whisking leads to interhemispheric decorrelation

Yael Oran [1,2], Yonatan Katz [1,2], Michael Sokoletsky[1], Katayun Cohen-Kashi Malina[1] & Ilan Lampl [1✉]

Interhemispheric correlation between homotopic areas is a major hallmark of cortical physiology and is believed to emerge through the corpus callosum. However, how interhemispheric correlations and corpus callosum activity are affected by behavioral states remains unknown. We performed laminar extracellular and intracellular recordings simultaneously from both barrel cortices in awake mice. We find robust interhemispheric correlations of both spiking and synaptic activities that are reduced during whisking compared to quiet wakefulness. Accordingly, optogenetic inactivation of one hemisphere reveals that interhemispheric coupling occurs only during quiet wakefulness, and chemogenetic inactivation of callosal terminals reduces interhemispheric correlation especially during quiet wakefulness. Moreover, in contrast to the generally elevated firing rate observed during whisking epochs, we find a marked decrease in the activity of imaged callosal fibers. Our results indicate that the reduction in interhemispheric coupling and correlations during active behavior reflects the specific reduction in the activity of callosal neurons.

[1] Department of Neurobiology, Weizmann Institute of Science, Rehovot, Israel. [2]These authors contributed equally: Yael Oran, Yonatan Katz. ✉email: Ilan.Lampl@weizmann.ac.il

Correlations in neuronal activity are a major hallmark of mammalian cortical physiology, first reported more than a hundred years ago[1]. Such correlations are believed to increase the saliency of information processing[2–4], but may also reduce the capacity to carry information[5–8]. While correlated activity in nearby cells is expected due to common inputs, it is less trivial between distant areas. One striking example of long-distance correlations is that between homotopic cortical areas across the two hemispheres[9–12]. When one area of the cortex becomes active, it is generally accompanied by corresponding activity in the homotopic region in the other hemisphere. Such correlations have been extensively documented in both animals[13–15] and humans[16–19].

Another hallmark of mammalian cortical physiology is the appearance of slow fluctuations in cortical activity over time, a phenomenon which is suggested to arise from shifts in brain state[18]. These fluctuations are accompanied by changes in the correlation structure of local neural activity. Specifically, studies in awake mice revealed that the cortex is decorrelated during bouts of exploratory behavior, such as whisking[20–22] and running[23–25], compared to quiet wakefulness periods.

The corpus callosum is an extensive fiber bundle in placental mammals that connects the two cerebral hemispheres. It is commonly suggested that the corpus callosum mediates interhemispheric correlations. Indeed, callosal lesion studies have mostly shown a reduction in interhemispheric correlations, both for sensory-evoked as well as ongoing activity[13,26]. Low interhemispheric correlations during ongoing activity were also reported in awake acallosal mice (transgenic mice line lacking the corpus callosum)[14]. However, these studies have not explored how interhemispheric correlations are affected by behavioral states in awake animals. Additionally, the invasive method of severing the corpus callosum used in these studies is not spatially and temporally precise and thus could affect interhemispheric correlations indirectly, such as by compensatory mechanisms. Finally, although abundant research was done on suprathreshold interhemispheric correlations, it remains unknown how such correlations are represented in subthreshold activity in awake animals.

Here we studied the mechanisms of interhemispheric correlations and their dependence on behavioral state in the somatosensory cortex of awake mice. We demonstrate that interhemispheric correlations are manifested both at the suprathreshold (action potentials) and subthreshold (membrane potential) levels and are modulated by behavioral state. Using optogenetics and chemogenetics, we show that state-dependent changes in interhemispheric coupling and correlations causally depend on callosal fibers activity. Finally, using two-photon calcium imaging we demonstrate that spontaneous callosal activity is modulated by behavioral state, and in an opposite manner to the local population. Our results provide important insight to the causal role of the corpus callosum in mediating interhemispheric communication, as well as its dependence on the behavioral state of the animal.

## Results

**Interhemispheric spike correlations in awake mice decrease during active behavioral states**. We first studied how whisking modulates the firing correlations across the two primary somatosensory cortices (S1–S1) of awake mice. Since rodents move their whiskers on both sides in a coordinated fashion during free-air whisking (Supplementary Fig. 1)[27–30], S1–S1 correlations could reflect shared whisking signals ascending from the thalamus. However, because local correlations are weaker during active states[22], usually characterized by whisking and locomotion,

interhemispheric correlations may follow this change and thus decrease during whisking.

To distinguish between these two possibilities, we simultaneously recorded neuronal activity from the two barrel cortices using laminar probes in awake head-fixed mice ($n = 141$ neurons, $n = 6$ mice) while monitoring whisking with a video camera to identify quiet wakefulness and active states (Fig. 1a, b, see Methods). Because we found that interhemispheric correlations were not related to the barrel maps (Supplementary Fig. 2), laminar probes were inserted without targeting specific barrel columns.

We automatically segmented each recording session to quiet wakefulness and whisking epochs (Fig. 1b, see also Supplementary Fig. 3 showing the percentage of whisking and quiet wakefulness epochs, see Methods). Firing rates were significantly higher during whisking epochs compared to quiet wakefulness, with 91% of neurons showing higher firing rates during whisking (Fig. 1c, $6.23 \pm 0.63$ Hz during quiet wakefulness, $11.70 \pm 0.96$ Hz during whisking, spikes/s, mean ± SEM; $p < 0.001$, $n = 141$ neurons, two-sided Wilcoxon Sign Rank Test [WSRT]). To examine the correlations within and between the two hemispheres, we calculated the spike-triggered average (STA) between all combinations of cell pairs (see Methods). Importantly, the shuffled STAs were subtracted to reduce the contribution of co-modulation of neuronal firing to the measured correlations[31]. Whisking was accompanied by a drastic reduction in the local correlations (Fig. 1d, 74.5% reduction relative to baseline, $p < 0.001$, $n = 850$ pair combination, 6 animals, WSRT) and a smaller yet significant reduction of interhemispheric correlations (Fig. 1e, 12.2% reduction, $p < 0.001$, $n = 800$ pair combinations, WSRT). A further examination of the distribution of interhemispheric correlations using a modulation index (MI, Methods), which quantifies the change of STA peak between the states, revealed significantly higher interhemispheric correlations during quiet wakefulness epochs (Fig. 1f, MI $= -0.1218$, $p < 0.001$, $n = 800$, WSRT). To observe pairwise correlations across the two hemispheres we present the entire population after sorting all based on the peak correlation across the two states (Fig. 1gi-ii). We found a change in the distribution of interhemispheric correlations, which manifested for most pairs as a marked reduction in STA magnitude during whisking epochs (Fig. 1gi-ii, entire population data sorted by mean correlation strength. giii, superimposed distributions of data in Fig, 1gi-ii, Supplementary Fig. 4). Note also that whereas during quiet wakefulness a small number of STAs exhibited negative values, they were almost absent during whisking epochs. Hence, we conclude that although whisking leads to an increase in mean population firing rate, it is accompanied by a marked reduction in the magnitude of interhemispheric correlations.

**Interhemispheric membrane-potential correlations in awake mice are also state-dependent**. To investigate how interhemispheric correlations are manifested at the subthreshold level we performed paired whole-cell patch-clamp recordings in L2/3 or L4 simultaneously from both right and left barrel cortices (S1–S1) in head-fixed awake mice freely able to walk on a treadmill (Fig. 2a; $n = 9$ pairs from 6 animals, mean depth of recording $346 \pm 107$ μm). In this set of experiments, whisking was tracked using a noncontact IR detector (see Methods).

Two examples of paired recordings are shown in Fig. 2b, f, in which a 20 second epoch of simultaneous membrane-potential recordings of left and right hemisphere neurons are shown together, demonstrating slow and fast correlated subthreshold activities. The slow component is observed in the raw data as a slow depolarization for both neurons and the fast component is

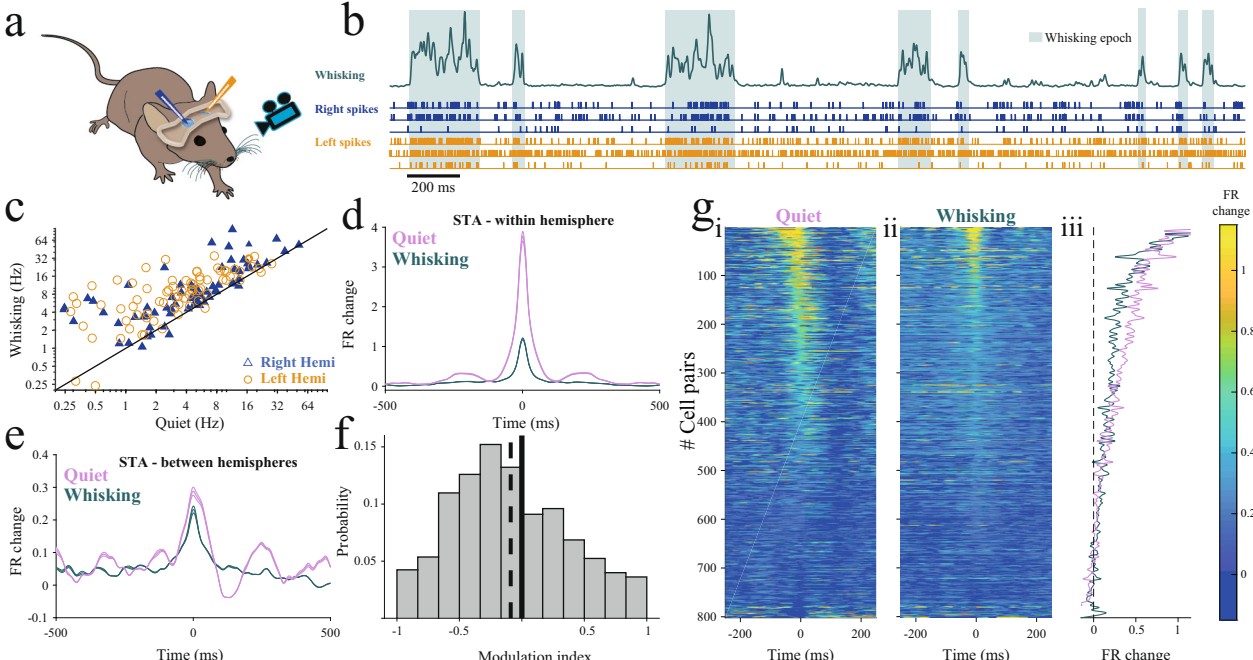

**Fig. 1 Interhemispheric correlations are reduced during whisking. a** Schematic diagram of the recording session. **b** An example of an extracted whisking signal and the associated states. Automatically detected whisking epochs (Methods) are marked by teal background. Below are three examples of raster plots of simultaneously recorded units in the right (blue) and left (orange) hemispheres. **c** Mean firing rates during whisking compared to quiet epochs for all the recorded neurons ($p < 0.001$, $n = 141$ neurons, WSRT). **d** Population average of the spike-triggered averages (STA) calculated for all combinations of cell pairs within hemisphere during quiet and whisking states (mean (line) ± SEM (shaded), $p < 0.001$, $n = 850$ pair combinations, 6 animals, WSRT). **e** Population average of the spike-triggered averages, calculated for all combinations of cells pair across hemispheres during quiet and whisking states (mean (line) ± SEM (shaded), $p < 0.001$, $n = 800$ pair combinations, WSRT). For **d** and **e** shuffled spike-triggered averages were subtracted (see Methods). **f** Distribution of state modulation indices defined as the difference between peak STA activity during whisking epochs and quiet wakefulness divided by their sum, calculated for all combinations of cells pair across hemispheres (dashed line marks mean modulation index). **g**i–ii Right and left panel depict STAs between hemispheres showing all pairs during quiet wakefulness (i) and whisking (ii). Both panels are ordered identically, based on mean peak STAs obtained from both states. **g**iii Superposition of the distribution curves of the average peak for whisking (teal) and quiet wakefulness (pink) for all pairs.

observed in the simultaneous rapid synaptic depolarization ('bumps'). These components are also observed in the gross cross-correlation between the traces (Fig. 2c, g: left panels), showing a fast component (less than a second) on top of a slower seconds-long component. The contribution of the faster component to the subthreshold interhemispheric correlation was revealed after high-pass filtering the traces above 1 Hz before computing the cross-correlation (Fig. 2c, g: right panels; see correlations for all recorded pairs at Supplementary Fig. 5).

The whisking signal that is presented together with the membrane potentials (Fig. 2b, f, teal) demonstrates that cells tend to depolarize during whisking epochs (from $-65.6 \pm 2.3$ mV to $-61.4 \pm 2.9$ mV at active state, Fig. 2l, $n = 18$ neurons)[22]. Depolarization of the membrane potential during whisking could explain the averaged increase in firing rate we found using the laminar probes (in Fig. 1c) and the slow component in the cross-correlation described above (Fig. 2c, g: left panels). Importantly, we found that whisking modulated the fast component of the interhemispheric subthreshold correlation. This is observed from the reduction in the running (sliding) window Pearson's cross-correlation 'WCC' trace in Fig. 2b, f, computed after high-pass filtering the traces above 1 Hz. This WCC analysis reveals a decrease in interhemispheric correlation during whisking epochs. The two expanded portions of the recordings in Fig. 2d, h indeed exemplify this dependency. During quiet wakefulness, synaptic 'bumps', which are reminiscent of brief Up and Down-like synaptic activities, were highly correlated across the two hemispheres. This suggests that the interhemispheric correlations of Up and Down states previously shown in anesthetized animals[32] also

exist during wakefulness[33]. An additional and independent analysis supported our initial findings that interhemispheric correlations were lower during whisking epochs. First, we computed the noise-correlation for each state by subtracting from each trace the mean potential of all epochs (see Methods[34]). This was done to remove the contribution of slow co-modulation and slow depolarization or hyperpolarization of the membrane potential during state shifts. We found a marked reduction in the correlation from 0.43 during quiet wakefulness to 0.10 during whisking for the (Fig. 2e) example in b and from 0.09 to 0.05 in the example in f (Fig. 2i). These findings were further substantiated when examining the population data in Fig. 2j, k, in which cross-correlations, obtained with and without high-pass filtering are shown for a population of 9 pairs respectively (in both cases they were analyzed as noise-correlations).

In addition, we calculated the running window interhemispheric subthreshold cross-correlation (after high-pass filtering the data above 1 Hz) and then cross-correlated the resulted signal with the whisking envelope. That these two signals are anti-correlated is clear both when inspecting them by eye and when calculating the cross-correlation between them (Fig. 2m, n demonstrate a single example and Fig. 2o shows the averaged data across all pair recordings). These curves demonstrate a negative peak centered near zero lag (the change in Vm correlation preceded whisking by 250 ms, Fig. 2o inset).

Altogether, and in agreement with the extracellular recordings, our paired intracellular recordings from the two barrel cortices show the following: (1) interhemispheric subthreshold correlations exhibit slow and fast components and had significant

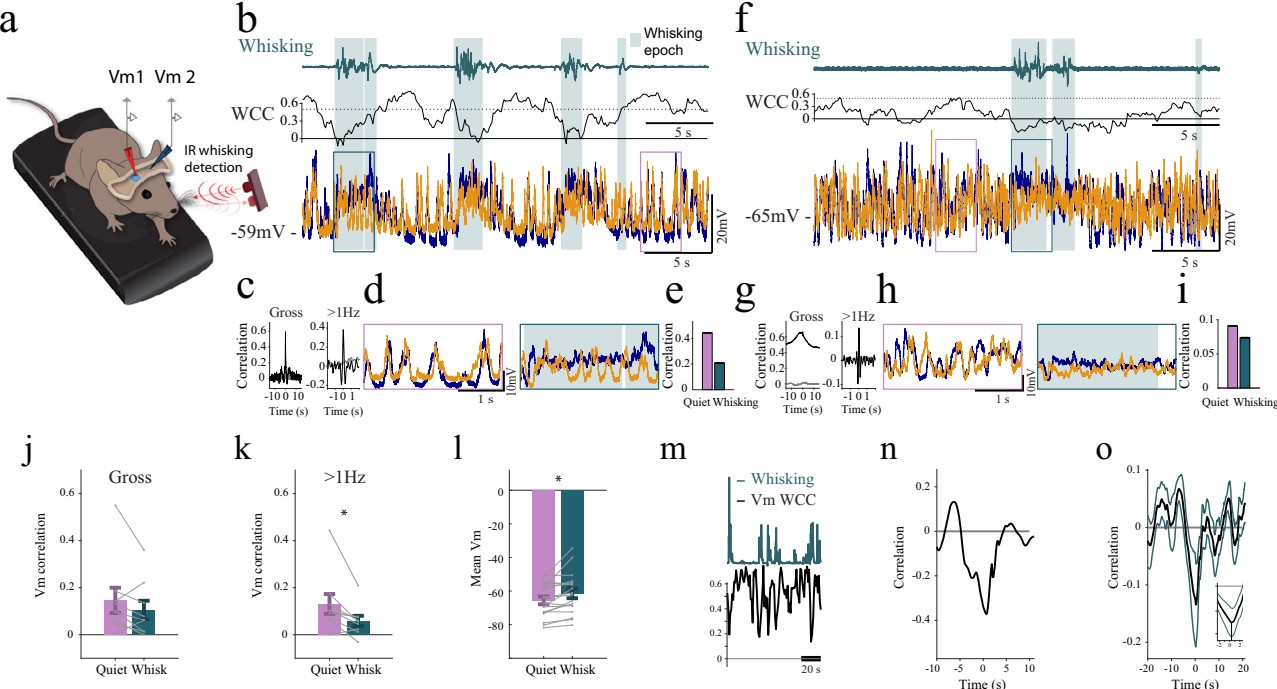

**Fig. 2 Decorrelation of interhemispheric subthreshold activities during whisking. a** Schematic diagram of paired whole-cell patch-clamp recordings simultaneously from both hemispheres. **b** An example of simultaneous dual whole-cell recordings from the right (orange) and left (blue) barrel cortices during quiet wakefulness and whisking epochs (teal background), along with the corresponding whisking activity (upper panel) and a running window membrane potential (Vm) cross-correlation (WCC, middle panel). **c** Cross-correlation of the membrane potential of the two neurons (shown in **b**) for raw data (left) and following filtering above 1 Hz (right). **d** Expanded portions of membrane potentials during quiet wakefulness (pink frame) and whisking epochs (teal frame), showing epoch of low and high correlated activities (left, right, respectively), **e** The mean subthreshold cross-correlations during quiet wakefulness and whisking epochs for the example in **b** (mean ± SEM, quiet $n = 82847$, whisking $n = 13154$, WSRT $P < 0.0001$, n refers to the number of steps in the running cross-correlation). **f–i** Same as **b–e** but for another example pair (for **i**, mean ± SEM, quiet $n = 542692$, whisking $n = 168311$, WSRT $P < 0.001$, n refers to the number of steps in the running cross-correlation). **j** Interhemispheric subthreshold correlation during quiet wakefulness and whisking epochs for all pairs (gray lines, individual pairs, mean ± SEM, $n = 9$ cell pairs, two-sided paired $t$-test $P = 0.155$). **k** Same analysis as in (**j**) following filtering the membrane potential above 1 Hz (mean ± SEM, $n = 9$ cell pairs, two-sided paired $t$-test $P = 0.018$). **l** Mean membrane potential during the two states (mean ± SEM, gray lines, individual pairs, $n = 9$ cell pairs, two-sided paired $t$-test $P = 0.007$). **m** Example of the whisking envelope (teal) with its corresponding running window Vm interhemispheric cross-correlation (black, 2 seconds sliding window of 10 ms bins). **n** Cross-correlation of whisking envelope with running window Vm cross-correlation of the example data in **m**. **o** Mean cross-correlation of the whisking envelope with running window Vm cross-correlation, obtained by averaging data from nine pairs (black line-mean, green lines ± SEM). The vertical line in the inset marks the peak of the cross-correlation, demonstrating that a reduction in Vm correlations precedes whisking by roughly 250 ms.

coherence up to 7 Hz (Supplementary Fig. 6), (2) whisking is accompanied by depolarization of the membrane potential of neurons in the upper cortical layers, which explains the slow component of interhemispheric cross-correlations, and 3) whisking drastically reduces the rapid synaptic interhemispheric correlations. Thus, like in the spiking activity, we conclude that even though right and left whisking movements were highly correlated (Supplementary Fig. 1), whisking is accompanied by a reduction in subthreshold interhemispheric correlations.

**Interhemispheric coupling depends on behavioral state.** Next, we sought to find the possible mechanism behind the observed state-dependence of interhemispheric correlations. On the one hand, this phenomenon might result from common inputs to both hemispheres, such as inputs from subcortical nuclei and neuromodulatory systems. On the other hand, it can emerge from direct or indirect recurrent connections between the two hemispheres. In the former case, where the two hemispheres are presumably not coupled (i.e., barrel cortex in each hemisphere does not influence the other) and the correlations emerge from common inputs, inactivation of one hemisphere should not influence the activity in the other. However, if the two hemispheres directly or indirectly influence each other, inactivation of

the barrel cortex in one hemisphere is likely to influence the activity in the barrel cortex of the other hemisphere. Moreover, if this is indeed the case, based on our observation that interhemispheric correlations depend on the behavioral state, we expect to find weaker coupling during whisking epochs (i.e., the effect of inactivating one hemisphere will have a more prominent influence during quiet wakefulness).

To test the strength of coupling between the two hemispheres and to find how it depends on the behavioral state, head-fixed GAD-ChR2 transgenic mice free to run on a treadmill were used to optogenetically inhibit the activity of the barrel cortex in one hemisphere while we intracellularly recorded from the other homotopic area. A paired intracellular recording example shows that ongoing subthreshold activity of the neuron in the inhibited hemisphere was completely suppressed during the optogenetic stimulation (Fig. 3a, b). Using the same line of mice, we previously showed that such light stimulation abolished firing across all cortical layers[35]. During the inactivation period, the activity in the opposite hemisphere was seemingly unaffected (Fig. 3b). However, additional analysis of the recorded neurons revealed a state-dependent effect. Sorting the trials based on the upper and lower 25th percentiles according to the whisking activity revealed a transient hyperpolarization in voltage activity

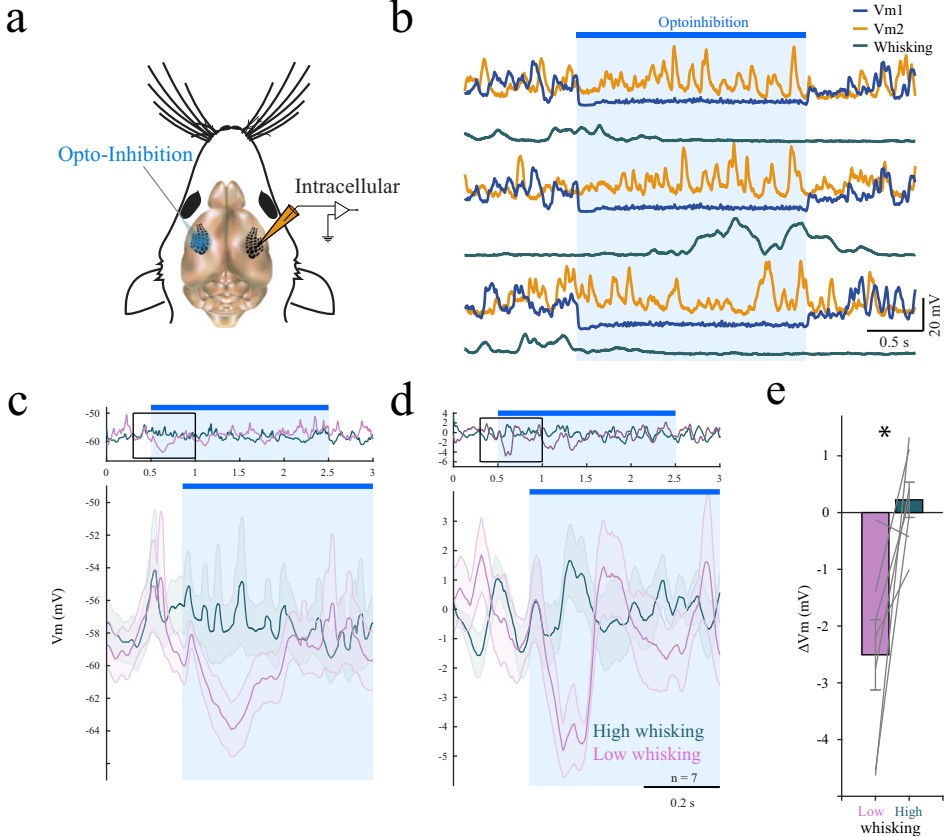

**Fig. 3 Interhemispheric coupling is reduced during whisking. a** Paired intracellular recordings in the right and left barrel cortices of head-fixed GAD-ChR2 mouse on a linear treadmill. **b** Example traces for paired intracellular recordings from right (orange) and left (blue) hemispheres shown together with whisking trace (teal). The left (blue) hemisphere was inactivated by light (blue bar), thus silencing ongoing activity in that hemisphere (Vm1, Vm2-membrane potential of the cells). **c** Example data form one neuron showing mean membrane potential for epochs of high whisking (teal) and low whisking (pink). During optogenetic inhibition (marked in blue) transient hyperpolarization was prominent only during low-whisking epochs (pink). **d** Population average of membrane-potential traces ($n = 7$ neurons) as in **c**. **e** Voltage change due to light for the low and high-whisking trials averaged across all pairs ($\Delta Vm$ = difference between mean membrane potential during first 200 ms following light inactivation and mean baseline Vm before light (800 ms), mean ± SEM, $p = 0.002$, two-sided WSRT, $n = 7$ neurons).

in the first couple hundred milliseconds after the onset of the light stimulation only during quiet wakefulness trials (Fig. 3c–e, hyperpolarization was significant only in quiet state, $-1.52 \pm 0.46$ mV, $p = 0.038$, WSRT). Hence, this result shows that the two hemispheres causally influence one another only during quiet wakefulness. Inactivation of one hemisphere led to a small but significant brief hyperpolarization in neurons in the other hemisphere only during quiet wakefulness, while no influence was found during whisking trails.

The effect of the light stimulation on the contralateral membrane potential could potentially result from a change in the whisking pattern, either as an arousing signal or due to blockade of whisking initiation signals from S1 to M1[36]. Illuminating the cortex in WT mice showed that the probability of whisking initiation was not changed by light (Supplementary Fig. 7A), indicating that the arousal state was not affected by the light stimulation itself. However, in agreement with Sreenivasan et al. 2016, inactivation of the barrel cortex reduced the probability of whisking initiation in GAD-ChR2 mice when inspected in the first 1000 milliseconds following light onset (Supplementary Fig. 7b). Yet, we found no measurable effect of the light stimulation on whisking pattern in the first 200 milliseconds following light onset in the low-whisking trials, where a clear transient hyperpolarization was observed in the membrane-potential activity (Fig. 3c–e). Hence, this transient

hyperpolarization cannot be explained by a change in whisking pattern. We therefore conclude that the coupling between the barrel cortices found only during quiet wakefulness is in agreement with our paired recordings, which showed higher interhemispheric correlations during this state.

**Unilateral inactivation of callosal transmission significantly reduces interhemispheric correlations.** The optogenetic experiment revealed state-dependent coupling between the two hemispheres, which grants further support to the previously observed state-dependence of interhemispheric correlations. However, it is not yet clear whether a direct or indirect pathway between the hemispheres mediates this effect. Since the corpus callosum directly connects the two somatosensory cortices in each hemisphere, we tested the role of the corpus callosum in the state-dependent correlation by manipulating its transmission. We first used the standard approach of severing the corpus callosum by performing a local suction of the region connecting the two barrel cortices. Simultaneous LFP recordings were obtained from both barrel cortices before and after the lesion. The interhemispheric correlations decreased after the lesion (Supplementary Fig. 8b). However, we found that this standard lesioning approach[37–40] also suppressed the overall cortical activity for a long period of time (>30 min, Supplementary Fig. 8f).

Although the mechanisms for this suppression are unclear, it could result from secondary effects of the lesion. To overcome this confound we used chemogenetics to unilaterally silence callosal synaptic transmission[41,42]. Following expression of hM4D in the left barrel cortex (via an injection of AAV2/5-hSyn-hM4D(Gi)-mCherry), we obtained simultaneous single and multi-unit recordings from the two barrel cortices of awake mice using laminar silicon probes in 6 mice. After 20 min of continuous recordings, CNO was topically applied to the non-injected right hemisphere (Fig. 4a) to specifically inactivate the terminals of callosal axons arising from the virally-transfected hemisphere.

STAs obtained from neurons across hemispheres (spikes in the left side were used as trigger for averaging the firing on the other side) revealed a small but significant reduction in interhemispheric correlations following the application of CNO during whisking epochs. However, a greater effect for the silencing of callosal transmission was found during quiet wakefulness. The magnitude of the STA (relative to STA before CNO application) was reduced by 50% during quiescent periods but only by 14% during whisking epochs (Fig. 4b, d, $p < 0.001$, WSRT, $n = 802$ pairs). This effect is also easily visible when examining the entire population data (Fig. 4c, e). These intensity plots show a clear reduction in magnitude for most STAs during quiet wakefulness and a much weaker effect during whisking epochs. Although CNO application affected the firing correlations between the two hemispheres, it had no significant effect on the mean firing rate in either hemisphere (Supplementary Fig. 9). In addition to the prominent reduction in the STA across hemispheres during quiet epochs, we observed a moderate reduction in that state also in the local STA measured from the activity in the left hemisphere, where hM4D was expressed (Supplementary Fig. 10). The reduction in local correlations perhaps reflects a loss of feedback to this hemisphere. Importantly, changes in interhemispheric and local correlations due to the application of CNO were not accompanied by any prominent changes in the whisking pattern (Supplementary Fig. 11). Hence, these results reveal that the effect of the corpus callosum on interhemispheric correlations depends greatly on the behavioral state of the animal and strongly suggest that the corpus callosum mediates the higher interhemispheric correlations and coupling observed during quiet wakefulness.

**Activity of callosal axons is state-dependent.** How is it that the corpus callosum plays a prominent causal role in mediating interhemispheric correlations in one behavioral state (quiet wakefulness) but not in another (whisking), as evident in the chemogenetic inactivation experiment? One explanation is that the activity of the corpus callosum itself is state-dependent. To test this, we injected GCaMP6s (AAV2/1-Syn-GCaMP6s-WPRE) in the barrel cortex (S1) of a single hemisphere and opened an imaging window above the barrel cortex of the contralateral hemisphere (Fig. 5a). This allowed us to view and record solely the fluorescence signal of axons crossing the hemisphere in layers 1 and layer 2/3 while monitoring whisking (Fig. 5a, b, histology was performed in 3 out of 5 animals). In the two example recordings shown in Fig. 5c, we observed lower activity during whisking epochs (see also Movie 1). This was further verified by computing the cross-correlation between the whisking envelope and the axonal calcium signal, showing a clear negative peak around zero lag for the two examples (Fig. 5d) and across the population of 45 axons (Fig. 5e, $n = 45$ from 5 mice). Control imaging experiments in awake mice expressing GFP in callosal axons (Supplementary Fig. 12) showed no correlation between GFP signals and whisking. This suggests that the negative

correlation observed between calcium signals and whisking signals was not caused by movement artifacts. To further ensure that this negative correlation is not due to movements, causing shifts in baseline calcium signal, we also analyzed only calcium signals above a certain threshold to capture high level of axonal activity (Fig. 5f, see Methods). Extraction of these thresholded calcium signals and subsequent segmentation of the data to whisking and quiet wakefulness epochs (as with the extracellular laminar probes, see Methods) revealed that during quiet wakefulness, the mean activity of the callosal axons was significantly higher by 29.46% compared to whisking epochs (Fig. 5g, $1.72 \pm 0.18\%$ for quiet wakefulness and $1.33 \pm 0.18\%$ during whisking). Note, however, that not all axons exhibited the same tendency to decrease their activity during whisking. Quantifying this effect using a modulation index revealed a mean negative state-dependent modulation of callosal axon activity (Fig. 5h, modulation index $= -0.1873$, $p = 0.0035$, WSRT, $n = 45$). Together, these results show that callosal activity is modulated by the behavioral state of the animal, with on average lower activity during whisking epochs. Interestingly, this contrasts with the increase in average firing rate observed during whisking in the local neuronal population (Fig. 1c). This increase in firing during whisking is also supported by additional imaging data from upper cortical layers obtained following expression of GCaMP6s in CaMKII neurons (Supplementary Fig. 13), showing a moderate positive correlation between whisking and local activity. Therefore, the activity of callosal neurons is modulated by behavior in an opposite manner to that of most neurons in the somatosensory cortex, suggesting that they form a separate functional sub-population. Altogether, these findings support our hypothesis that state-dependent corpus callosum activity leads to state-dependent interhemispheric communication.

**Discussion**

In this study, we examined the relationship between behavioral state and interhemispheric correlations. We showed that inter-hemispheric correlations between the left and right somatosensory cortices of head-fixed awake mice depend on whisking state. Simultaneous extracellular recordings from right and left hemispheres revealed that the majority of interhemispheric correlations were positive and decreased during whisking, despite the general elevation in firing rates. In agreement with these results, paired intracellular recordings showed that interhemispheric correlations also manifested at the subthreshold level, exhibiting higher synchronized synaptic activities during quiet wakefulness. Furthermore, optogenetically silencing the barrel cortex in one hemisphere influenced the other hemisphere only during quiet wakefulness, indicating that interhemispheric coupling is state-dependent. Unilateral chemogenetic silencing of callosal terminals resulted in a reduction in interhemispheric correlations, most prominently during quiet wakefulness. Finally, in contrast to the overall elevated firing rate of neurons in the barrel cortex during whisking, the calcium signals of callosal fibers were lower during this state, suggesting a differential modulation of their activity in an opposite direction to the majority of neurons in the somatosensory cortex.

**Local and interhemispheric decorrelations during whisking.** Synchrony and state-dependent modulation of cortical firing are two major hallmarks of cortical dynamics. In recent years, electrophysiological recordings from head-fixed awake rodents have been routinely accompanied by monitoring various behavioral signals, such as locomotion, whisking and pupil diameter, indicative of the awake brain states[24]. Whether or not the mean firing rate of cortical neurons increases during free whisking is

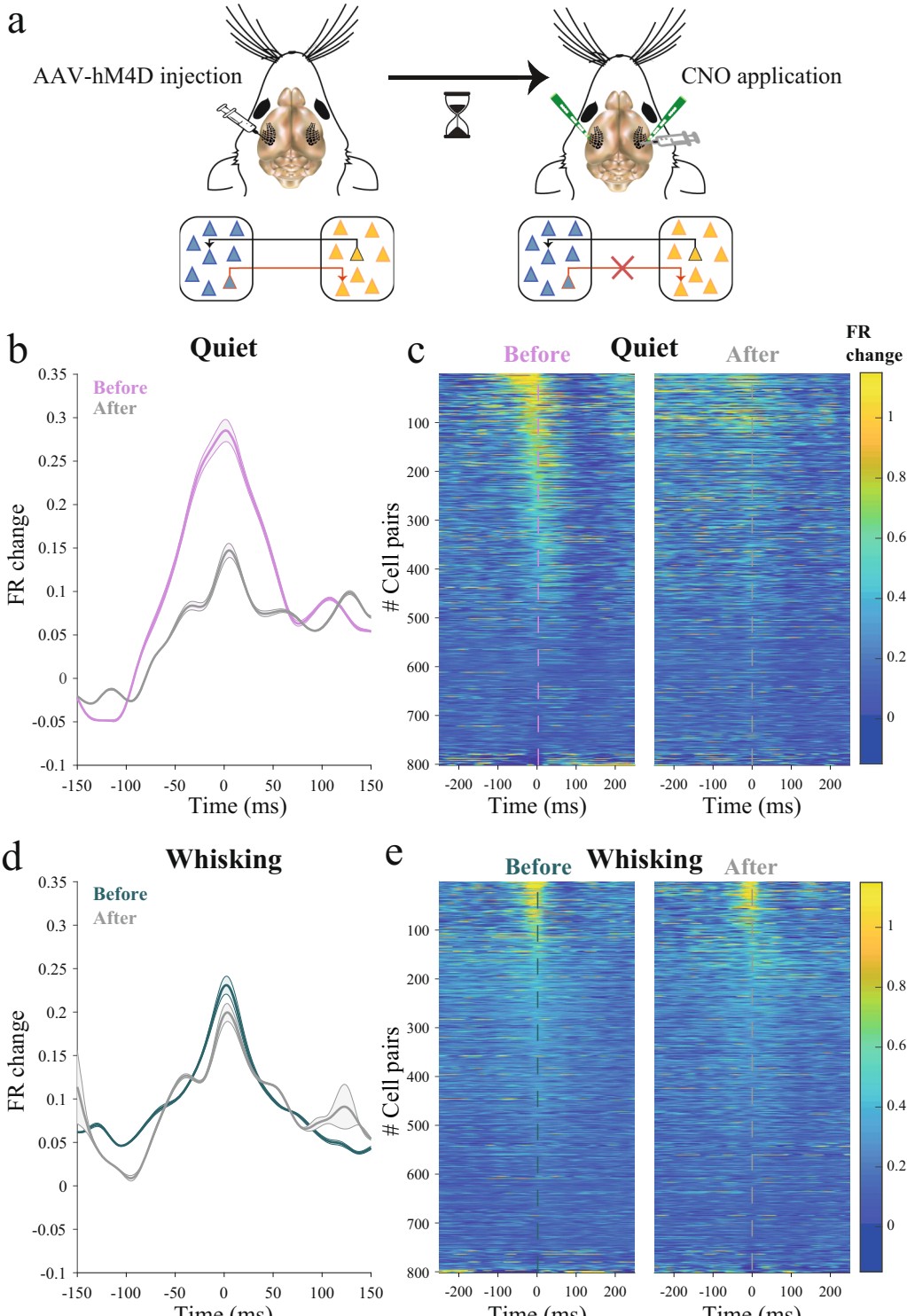

**Fig. 4 Unilateral silencing of callosal projections leads to a state-dependent reduction in interhemispheric correlations. a** Schematic diagram of the injection and experiment. hM4D was expressed in the left hemisphere and following expression extracellular recordings were made simultaneously from both barrel cortices. CNO was topically applied to the right barrel cortex, silencing callosal terminals arising from the left hemisphere. **b, d** Population average of spike trigger average (STA) between the hemispheres before and after application of CNO as calculated during quiet wakefulness (**b**) and whisking (**d**) (mean (line) ± SEM (shaded), $p < 0.001$, two-sided WSRT, $n = 802$ pairs). Shuffled STAs were subtracted (see Methods). **c, e** STAs between hemispheres before (right) and after (left) application of CNO, showing all pairs during quiet wakefulness (**c**) and whisking (**e**). Both panels are ordered identically, based on mean peak STAs obtained from both states.

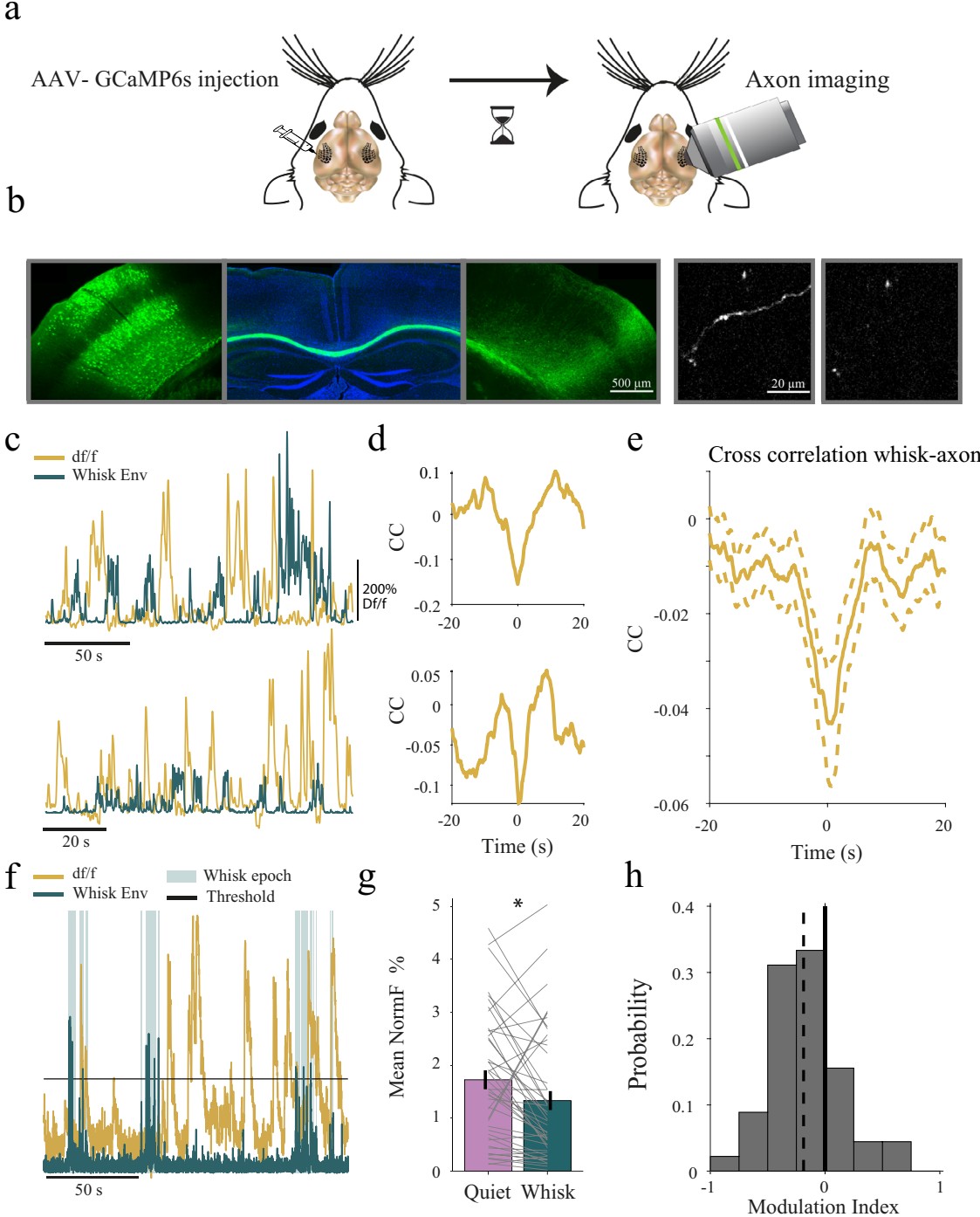

**Fig. 5 Activity of callosal axons is reduced during whisking. a** Schematic diagram of the injection and imaging procedures. **b** Histology of the injection (leftmost) and callosal projecting axons (second and third from the left), images were taken from the same slice with different exposure levels. The two right most panels show an active and non-active GCaMP6s-expressing axon (histology was performed in 3 out of 5 animals) **c** Two examples, from two different animals, of simultaneously recorded callosal axon calcium signals (df/f, orange) and the whisking envelope (dark green). **d** The cross-correlations (cc) between the callosal axon and whisking signals for the two examples in **c**. **e** Population average of the cross-correlation between callosal axon activity and whisking envelope (mean ± SEM, n = 45 axons). **f** An example of axonal calcium activity with a threshold demonstrating the exclusion of baseline variability. **g** Mean callosal axon activity (above threshold) during quiet wakefulness and whisking epochs. Each line represents the above threshold calcium signal from a single callosal projecting axon (mean ± SEM, p < 0.001, WSRT, n = 45 axons). **h** Histogram of the state modulation index, defined as the ratio of the difference between axonal activity during whisking and quiet wakefulness divided by their sum (dashed line marks the mean, modulation index = −0.1873, p = 0.0035, WSRT, n = 45 axons).

controversial. While some studies, both in freely moving and head-fixed animals, found little change in the population firing rate[21,43–46], others showed elevated firing[47,48]. Such discrepancies might be explained by differences in experimental parameters, such as recordings from freely moving versus head-fixed animals, habituation procedures, as well as differences in the cell types and cortical layer of the recorded neuronal population. While our extracellular recordings, demonstrating higher firing rate during whisking (Fig. 1c), are more consistent with the latter group of studies, our imaging data (Supplementary Fig. 13) suggests a more moderate elevation in neuronal activity during whisking. We hypothesize that the elevated cortical activity during whisking may reflect different processes, including: (1) increased firing rate of thalamic cells[46,49], (2) activation of different neuromodulatory axons, such as cholinergic and noradrenergic[50,51], which leads to increase of cortical firing[52–55], and (3) inputs from the motor cortex which modulate firing in S1, as was previously suggested[56,57]. The differences we found between the different recording methods within our own study, however, may reflect a possible bias for a specific cell population in each given method. Nevertheless, our results indicate that whisking suppresses the firing of callosally projecting neurons.

In contrast to the clear elevation in firing during whisking epochs, local correlations were reduced as revealed by computing the spike-triggered averages between neurons (Fig. 1d). In agreement with previous studies demonstrating a strong decorrelation of local activity during active behavioral states, we found a decrease of interhemispheric correlation during whisking[22,24,58,59]. However, as we discuss below, the mechanism that leads to the reduction in interhemispheric correlations appears to differ from that which governs local correlations.

**Decorrelation of subthreshold activity during whisking**. Our paired intracellular recordings revealed the underlying subthreshold correlates of the state-dependent changes in interhemispheric correlations. Surprisingly, our findings show that during quiet wakefulness, brief Up and Down states were highly correlated across the two hemispheres (Fig. 2d). Subthreshold interhemispheric correlations of slow membrane-potential oscillations, reminiscent of the activity we observed in our study, were previously revealed by Contreras and Steriade while using paired intracellular recordings[32]. However, the study of Contreras and Steriade was conducted in deeply anesthetized cats and so did not reveal the state-dependent nature of these correlations. Our study quantifies the synaptic interhemispheric correlations in awake head-fixed mice and explores how these correlations depend on behavioral state.

**Mechanisms of state-dependent interhemispheric correlations**. What are the possible mechanisms of the interhemispheric correlations that we found in awake mice? Interhemispheric correlations were suggested to arise from direct callosal projections. However, we will first discuss and consider the possibility that such correlations emerge due to other mechanisms.

Ascending whisking signals from the thalamus can enhance interhemispheric correlations. Since rodents move their whiskers on both sides in a coordinated fashion during free-air whisking (Supplementary Fig. 1)[27–30], one would expect to find higher interhemispheric correlation during whisking. However, this was not the case in our extracellular and intracellular recordings, suggesting that interhemispheric correlations are unlikely to result from common thalamic inputs. The two hemispheres can also be co-activated by the neuromodulatory system. For example, it has been shown that cholinergic axons that innervate the cortex increase their firing during active states characterized

by locomotion or whisking[51,60]. Either activating the cholinergic system or applying a cholinergic agonist directly shifts cortical dynamics from a synchronous to an asynchronous state[34,61,62]. Likewise, it was shown that noradrenergic neurons fire about a second before each brief synchronized depolarizing synaptic cortical event[63,64]. Hence, these systems can enhance the correlations between the two hemispheres. Finally, these correlations can also be shaped by other noncortical brain regions, such as the striatum, which exhibits bilateral vibrissa responses[65] and thus may play an important role in the state-dependent interhemispheric correlations.

Although we cannot rule out the possibility that interhemispheric correlations are driven by neuromodulation, the emergence of interhemispheric correlation due to such common inputs implies that inactivation of one hemisphere would not affect the other hemisphere. In contrast, when we inactivated the barrel cortex of one hemisphere, we found a significant change in the membrane potential of neurons in the other hemisphere (Fig. 3). Importantly, this change was observed in a manner expected from our paired extracellular interhemispheric recordings, which demonstrated a state-dependent effect on interhemispheric correlation. During whisking epochs, the silencing of one hemisphere did not affect the activity in the other hemisphere, whereas during quiet wakefulness, a significant transient hyperpolarization was recorded. Previous studies established that callosal inputs make direct excitatory connections onto cortical pyramidal cells[66,67] and drive disynaptic feedforward inhibition onto local GABAergic interneurons[68,69]. Therefore, this transient hyperpolarization is likely to reflect the withdrawal of direct excitatory callosal input onto specific neurons which is quickly compensated by local network effects. This possibility was also suggested by a recent study by Slater and Issacson, which demonstrated a transient reduction in the firing of neurons in the auditory cortex when a similar optogenetic approach was used to silence firing in the contralateral side[70]. Likewise, cooling the barrel cortex in one hemisphere leads to a reduction in firing rate of neurons in the contralateral barrel cortex[71]. This transient hyperpolarization could also result from direct activation of inhibitory fibers projecting to the other hemisphere[72]. However, we failed to observe any transient hyperpolarization in response to light during whisking epochs, a state in which inhibition should be observed more easily due to a more depolarized membrane potential and thus further away from the inhibitory reversal potential. The study of Slater and Issacson (2020) also showed a delayed excitation following contralateral silencing. The lack of a delayed depolarization in our experiments following contralateral silencing perhaps reflects differences between the auditory and somatosensory cortices.

Overall, our results of positive interhemispheric correlations together with hyperpolarization following unilateral inhibition suggest that interhemispheric inhibition during ongoing activity is negligible under our experimental conditions. Interhemispheric inhibition was reported in several studies which argued that excitatory callosal fibers exert disynaptic feedforward inhibition through local GABAergic interneurons[68,69,73,74]. It is possible, however, that interhemispheric inhibition is recruited during whisker-touch events, which were not studied in our experiments. In summary, we suggest that excitatory coupling is the major driver of interhemispheric correlations during ongoing activity and that this coupling becomes weaker during whisking epochs.

The vital role of the corpus callosum in the emergence of interhemispheric correlations in animals was studied traditionally using partial or complete callosotomies. These studies demonstrate a reduction in ongoing and sensory-evoked interhemispheric correlations in the visual cortical areas of cats[13,15]. In another group of studies, it was shown that interhemispheric

correlations are weaker in genetically acallosal mice[14,75]. However, the weaker interhemispheric correlations in these studies may also reflect the potential involvement of compensatory or developmental mechanisms. Moreover, in our study we found a prolonged suppression in cortical activity following local lesioning of the corpus callosum (Supplementary Fig. 8), and therefore could not compare the signals before and after the ablation of the corpus callosum for the same neurons.

Our chemogenetic silencing of callosal terminals strongly suggests that the corpus callosum mediates the state-dependence of interhemispheric correlations. Our findings show that following the silencing of callosal terminals, a greater reduction in interhemispheric correlations was observed during quiet wakefulness compared to whisking epochs. A priori, because silencing was unilateral, we expected to find a clear change in the symmetry of the STAs. However, it is far from trivial which form this change should take. Moreover, it is known that in addition to callosal projections between homotopic barrel cortices, denser projections are found onto more lateral somatosensory areas, near the border between S1 and S2[66,76,77]. This raises the possibility that our chemogenetic silencing would result in an even a greater effect if CNO was applied to more lateral areas, blocking any contralateral inputs to these areas, which can propagate back to the recorded barrel cortex. In addition, it is possible that callosal projections between the two barrel cortices are specific based on the barrel maps. Yet, using paired LFP recordings from homotopic and non-homotopic barrel columns (Supplementary Fig. 2) we failed to find a fine structure in the interhemispheric correlations during ongoing and whisking periods. Surprisingly, the application of CNO also led to a reduction in local correlations in the hemisphere in which it was injected. The underlying mechanisms for this change are not clear and might reflect a loss of feedback inputs from the hemisphere in which the callosal fibers were inactivated. Future experimental and computational studies of interhemispheric coupling and synchrony may shed more light on the expected dynamics following such manipulations.

**State-dependent modulation of callosal activity.** Our calcium imaging experiments show that the activity of callosal fibers is modulated by the brain state in a manner that is in agreement with the other experiments presented in this study. For most callosal axons, calcium signals during quiet wakefulness were significantly higher than those recorded during whisking epochs. This was verified using two independent analysis methods, together with a control experiment in which we imaged axonal GFP signals (Supplementary Fig. 12) showing the calcium signals were unaffected by movement artifacts. The mechanisms underlying the reduction of callosal activity during whisking are yet unknown and can reflect the activity of inhibitory inputs. Since whisking has diverse effects on the firing rate of excitatory and inhibitory cells in the barrel cortex[46], callosally projecting neurons could be modulated in a specific manner. Indeed, when compared to other long-range projecting pyramidal cells, callosally projecting cortical neurons are intrinsically less excitable and exhibit a weaker response to contralateral callosal inputs[78,79]. Taken together, our results position the corpus callosum not only as a key player in mediating interhemispheric correlations but also as enabling the state-dependent characteristics of these correlations.

**Possible functional roles of interhemispheric correlations.** The roles of interhemispheric correlations in high brain functions, such as perception, learning and decision-making remain to be explored. Previous studies suggest that the prominent correlated activity observed during quiet wakefulness and sleep, characterizing the default state of the cortex, may serve some role in the maintenance of excitability[12,14,18,80]. More specifically, we suggest that interhemispheric correlations during quiet wakefulness enhance activity-dependent mechanisms that maintain the homoeostasis of synaptic and intrinsic excitability, and thus equalizing the sensitivity of homotopic cortical areas to sensory processing and other high brain functions. This process might be needed since during exploration sensory circuits on both sides are not necessarily activated at either the same rate or intensity, potentially leading to a drift in their set-points of sensitivity.

Whether or not state-dependent callosal activity and its contribution to interhemispheric correlations are also present in freely moving animals remains to be explored in future studies. The functional role of the reduction in interhemispheric correlations during whisking might be related to the proposed role of local decorrelation for enhancing sensory representation in the cortex during active states[81,82]. In this study, we did not explore how sensory stimuli are represented in both hemispheres. However, we speculate that although mean interhemispheric correlations decrease during active states, bilateral tactile inputs due to touch signals may boost the activity of a specific subset of callosal fibers, leading to spatially-restricted interhemispheric correlations. Enhancement of the correlations in a subset of neurons across the hemispheres may then enhance bilateral perception. It was argued that similar to local correlations during sensory perception, interhemispheric correlations may serve as a potential mechanism for binding different features of an object into one entity[13]. For example, mice spend a significant portion of their lives navigating tunnels underground, where interhemispheric correlations may allow them to perceive the tunnels as unified wholes.

In summary, our results provide direct evidence that state-dependent changes in the activity of the corpus callosum alter the communication between the two hemispheres. We hope it will pave the way for future studies in which interhemispheric communication are examined during naturalistic bilateral behavior.

## Methods

**Animals.** All experiments were conducted according to the Weizmann Institute Institutional Animal Care and Use Committee. In this study, a total of 28 mice were used. Mice aged 7–15 weeks had access to food and water ad libitum and were maintained under controlled humidity and temperature (45–65%, 22 ± 2°C, respectively) at a 12-h light/dark cycle. Paired intracellular recordings were conducted in GAD2-IRES-Cre mice (JAX, stock #010802) crossed with Ai32 (Stock #012569). Optogenetic silencing experiments were conducted in 4 animals of the same mouse line as above and 3 C57BL/6 animals. For axonal imaging experiments, 5 C57BL/6 animals were injected with AAV-Syn-GCaMP6s-WPRE-SV40 and for unilateral chemogenetic silencing experiments, 6 mice were injected with AAV2/5-hSyn-hM4D(Gi)-mCherry. Control imaging experiments of GFP expression in callosal axons were performed in two C57BL/6 animals injected with AAV-CaMKIIa-EGFP (Addgene, #50469-AAV5). Corpus callosum lesion experiment were made in 8 C57BL/6 animals.

**Surgical preparation.** For all our experiments animals were prepared for recordings by placing them in a stereotaxic frame under isoflurane anesthesia. Body temperature was maintained at ~37 °C using a regulated heating blanket and a thermal probe. Synthomycine 5% cream was used to cover the mouse eyes during the surgery. After fur removal and skin disinfection with iodine and ethanol, analgesics were given (Buprenorfine 0.1 mg/Kg, Carprofene 5 mg/Kg, I.P). An incision was made to the skin with a scalpel. Animals recovered for 4 days following surgery. In all the experiments the animals were mounted with custom-made 3D printed headbar attached to the skull (RelyX 3 M) to ensure head fixation during the experiment. These headbars were customized for awake electrophysiology and imaging.

**Viral injections.** During surgery, a craniotomy of ~0.75 mm was made above the right barrel cortex (AP: 1.3, ML: 3.2). A glass pipette (75-micron diameter) was back-filled with mineral oil and then front-filled with 1000 nL of virus (AAV2/5-hSyn-hM4D(Gi)-mCherry, AAV-Syn-GCaMP6s-WPRE-SV40 or

AAV-CaMKIIa-EGFP), which was injected in three to four different depths to ensure expression throughout the cortical layers. The pipette was advanced in the brain and the virus was unloaded slowly (30 nL/10 min) in each depth. The pipette was not moved between injections for ~2 min.

For calcium imaging of midline-crossing callosal fibers, five animals were injected with AAV-Syn-GCaMP6s-WPRE-SV40 (as described before) in the right barrel cortex. Afterwards a circular cranial window of 4 mm was made using a dental drill above the left barrel cortex (uninjected hemisphere). The craniotomy was then sealed with a 4 mm coverslip (150-micron thickness) and cyanoacrylate (Krazy Glue).

**Intrinsic imaging**. Animals previously mounted with headbars were anesthetized using isoflurane (2%) supplemented with chlorprothixene (2 mg\kg) and mounted in a stereotaxic frame. A large area (~1.5 mm radius) above the barrel cortex (centered at 3.3 mm lateral, 1.2 mm caudal from bregma) was thinned and soaked with mineral oil (M8410, Sigma–Aldrich) to make the bone more transparent. Following thinning, isoflurane concentration was reduced to 0.5% to increase neuronal activity. An image of blood vessels was acquired using green light (530 nm) and intrinsic signals were collected under red light (630 nm) using macroscope lens (X1 magnification, 50 mm tandem configuration) and mounted camera (MV-D1024E-40-CL-12, Photonfocus, Switzerland).

Imaging was acquired at 24 Hz under red light illumination during 12 s trials containing 2 s of 10 Hz whisker stimulation controlled by custom-written LabVIEW 2018 (National Instruments, Austin, TX, USA) program. The signals were analyzed using a custom-written MATLAB code (R2018a and R2019b; MathWorks). During the procedure, whiskers A1 and D1 on both hemispheres were mapped for targeted LFP recordings (Supplementary Fig. 2).

**LFP recordings**. Custom-made bundles for LFP recordings were built using formvar-insulated nichrome wire (#761500, AM systems, WA, USA). The recording bundles were coupled to an Intan RHD2132 headstage and data was sampled at 30 kHz using Open Ephys (v0.4.4)[83]. Each bundle had a reference channel and two signal channels inserted 550 µm beneath the dura using a micromanipulator (MX7600; Siskiyou, Grants Pass, OR) into miniature craniotomies over the A1 and D1 barrels. Data was further band-passed to 0.1–200 Hz and analyzed using custom MATLAB code (R2018a and R2019b; MathWorks).

**Lesion procedure**. Animals previously mounted with headbars were anesthetized using isoflurane (1-2%), a 0.5 mm × 0.5 mm craniotomy between the two barrel cortices (on the midline, 1.3 mm caudal) was made. A glass pipette with a broken tip connected to a narrow suctioning pipe was inserted 1.4 mm deep using a motorized manipulator (MX7600; Siskiyou, Grants Pass, OR). After 5 min that the pipette was in place (to allow the brain to return to normal position), the baseline LFP was recorded for about 20 min. Following this recording, suction was turned on for a couple of seconds. Five minutes following the end of suctioning the LFP was again recorded for 20 min.

**Whisking detection**

*Whisking detection via camera*. In all experiments other than paired intracellular recordings, whisking traces were extracted from facial videos. To extract the whisking behavior from the video, an ROI in the central part of the whisker pad was selected using a custom LabVIEW 2018 graphical user interface, which computed the squared difference in intensity between the pixels of each pair of consecutive frames. The whisking trace was defined as the sum of the squared differences for all pixels divided by the number of pixels within the ROI, i.e., the mean squared error between frames. The signal was smoothed with a Gaussian window (σ = 200 ms) and whisking episodes were detected using a threshold (1.5 times the upper 15th percentile of the whisking signal). Whisking episodes were identified as high-amplitude changes in the signal crossing the threshold, with at least 500 ms in duration. Minimum times between two episodes were set to at least 500 ms. Episodes crossing the threshold but not meeting the duration and separation requirements were ignored. The rest of the data were defined as quiet wakefulness with a 100 ms buffer time between whisking and quiet wakefulness.

*Whisking detection via reflective sensor*. During paired intracellular recordings, whisking epochs were detected by IR reflective sensor (HOA1405, Honeywell, NC, USA). It was positioned above the whiskers, pointing toward the whiskers' pad at a distance of 5 mm. The signal was acquired at 10 KHz using custom-written Lab-VIEW 2018 software. Whisking epochs were defined as amplitudes bigger than 1.5 times of the envelope signal's STD (MATLAB, R2018a and R2019b; MathWorks). Threshold crossing epochs with inter-epoch-interval smaller than 200 ms were binned together and defined as whisking epochs.

In Fig. 3, trials were sorted according to the standard deviation of whisking level during the trial (start to 1.5 s). Hence high-whisking trials consist of trials during which the animal spent a significant amount of time whisking, while in low-whisking trials the animal spent little to no time whisking.

**Extracellular recordings in awake mice**. Experiments began following at least 4 days of recovery from headbar mounting. Mice were anesthetized (~1.5% isoflurane), head-fixed and injected with carprofen (5 mg/kg). Under anesthesia a craniotomy (<0.5 mm radius) was performed over the two barrel cortices. The craniotomies were than covered with warm agar and sealed using silicone elastomer (Body Double™ Fast Set, Smooth-On, Inc., Macungie, PA). Thereafter, the animals were returned to their home-cage to recover for 2 h. Following recovery, the animals were lightly anesthetized with isoflurane, head-fixed, and the silicone elastomer was removed. A silicon laminar probe (NeuroNexus - Ax16-10mm-100-500-177) was inserted to each hemisphere using motorized manipulators (MX7600; Siskiyou, Grants Pass, OR). The silicon probes were coupled to an Intan RHD2132 headstage and the signals were sampled at 30 kHz using Open Ephys. The recording session started 15 minutes after the anesthesia was discontinued and the animals were fully awake. For chemogemetic silencing of callosal terminals, baseline recordings of at least 30 min were performed before applying CNO. Each experimental session began with recordings in the naive state, which was followed by a topical application of CNO (Clozapine N-oxide dihydrochloride, Tocris Bioscience) dissolved in artificial CSF to a final concentration of 1 mM. The CNO was applied to the barrel cortex on the contralateral hemisphere to the one in which the AAV-hM4D was injected in order to target callosal terminals only.

**Spike sorting and spike-triggered average**. The recorded data were preprocessed offline to multi-unit and single-unit spike trains (wave_clus toolbox for MATLAB as described here[84]). Neurons presenting unstable waveforms or firing patterns were excluded from the database. All the offline analyses were performed using custom-written MATLAB code (R2018a and R2019b; MathWorks).

Normalized spike-triggered averages were calculated for whisking/quiet wakefulness. These averages were normalized by the target cell mean firing rate (by dividing the resulting STA by the mean firing rate of the target cell). We calculated a shuffled STA by matching each of the whisking (or quiet wakefulness) epochs of the triggering unit with a different epoch of similar length in the target unit, and then calculated their spike-triggered average. We than subtracted the shuffled STA from the STA. This allowed us to remove correlations that are induced due to average modulations in firing rates following transitions to whisking.

**Intracellular recording in awake mice**. Following a recovery period of at least four days after headbar implantation, animals were anesthetized (~1.5% isoflurane), head-fixed and injected with carprofen (5 mg/kg). Under anesthesia craniotomies (<0.3 mm radius) were performed over the barrel cortices. Following that, the craniotomies were covered with warm agar (2%) and sealed using silicone elastomer (Body Double™ Fast Set, Smooth-On, Inc., Macungie, PA). Thereafter, the animal was returned to its home-cage to recover for 2 h. After recovery animals were anesthetized and head-fixed on a linear treadmill, the silicone elastomer was removed, then the anesthesia was discontinued, and the recording session started 30 min later when the animals were fully awake.

Patch electrodes were pulled using a dual-stage glass micropipette puller (PC-10, Narishige, Japan) and filled with an intracellular solution containing the following (in mm): 136 K-gluconate, 10 KCl, 5 NaCl, 10 HEPES, 1 MgATP, 0.3 NaGTP, and 10 phosphocreatine (310 mOsm). The resistance of the patch electrode was 5–10 MΩ.

The recording electrodes were targeted using a motorized manipulator (MX7600, Siskiyou).

Pipettes on both hemispheres containing high pressure were manually advanced at a fast rate (0.2–0.6 mm/s) down to 250 µm below the pial surface. Then, speed was reduced to approximately 2 µm/s, while searching for cells on one side based on pipette resistance. Following a successful whole-cell patch on one side, the other electrode was advanced to search for another cell on the other side, which was then also patched. Signals were amplified using MultiClamp 700B (Molecular Devices, San Jose, CA) and low passed at 4 kHz before being digitized at 10 kHz. No correction for junction potential was made (~12 mV).

**Subthreshold interhemispheric correlations**. In Fig. 2, different calculations were used for the evaluation of the relationships between behavioral state and inter-hemispheric subthreshold correlations:

In the first method, we used a moving window (2000 ms, steps of 10 ms) to calculate the Pearson correlation between the membrane potential of the two cells (using the 'xcorr' MATLAB R2018a function, setting normalization to 'coeff'). The vector resulting from this running cross-correlation was then cross-correlated with the simultaneously recorded whisking envelope (see section 'Whisking detection using reflector sensor').

In the second method, we computed the average cross-correlation between the cells for each state. Whisking epochs were identified from the whisking envelope by detecting onset and termination of whisking epochs using threshold as explained above. Quiet wakefulness epochs were defined as all times, which were not whisking epochs. This method provided epochs of variable durations that should be considered when computing the average cross-correlation as we further describe below. In addition, for each paired recording we either used the raw membrane-potential recordings or high-pass filtered the entire recordings above 1 Hz (using a Butterworth filter) to remove the contribution of slow changes in the membrane-

potential signals. To further subtract the effect of shifts in state, we calculated the 'noise-correlation' for each type of epoch (Meir et al. 2018). This was done by subtracting the average membrane potentials of the cells for each epoch type. We considered epochs with a minimal duration of 120 ms, and averaged only overlapping lengths of data across epochs. This was done by creating a set of membrane-potential vectors, each with a length corresponding to the longest detected epoch while setting NaN values between the ending of the original epoch and the longest one. These vectors were then averaged using the 'nanmean' MATLAB function to generate the mean membrane potential for a given type of epoch. The averaged mean membrane potential was removed from each original epoch (keeping its original length), thus removing the 'signal correlation' that emerges due to the onset of epochs. Then, for each epoch, we computed the cross-correlation between the two cells and averaged them in a weighted manner depending on their duration. The weighted averaging of these cross-correlations is based on their length since longer ones were calculated from more data. Thus, if n (i) is the number of points of an epoch i and c is the cross-correlation vector for this epoch (truncated at ±120 ms, the minimal epoch length) we computed the mean cross-correlation $\bar{C}$ as follows:

$$\bar{C} = \frac{\sum\limits_{i=1}^{n} n(i)*c(i)}{\sum_{i=1}^{n} n(i)} \qquad (1)$$

In summary, the first type of analysis in which running cross-correlation is used allowed us to see how interhemispheric correlations vary over time. However, this analysis suffers from the contribution of signal correlation, which we accounted for in the second type of analysis.

**Calcium imaging experiments**. Experiments began following calcium expression and a recovery period of at least 5 days from the surgery in which a coverslip window was mounted (see the section 'Viral injections').

During imaging, mice were head-restrained on a treadmill in a dark environment. Two-photon imaging was performed with Ultima, Bruker microscope. The imaging frame rates were 15–30 Hz and 920 excitation was used (x20 XLUMPlanFLN, Olympus). The whisking activity was monitored throughout the experiment using a webcam (ELP) recording at 30 Hz and illuminating with a 850-mm infrared LED light. Custom LabVIEW 2018 software was used to acquire whisking activity from video as described above. Only imaging fields showing at least a few micrometers of axons were analyzed.

**Preprocessing of calcium imaging data**. Regions of interest containing axons, or control regions with auto-fluorescent bulbous shape were segmented by hand with custom LabVIEW 2018 software. Imaging of GCaMP6s fluorescence in callosal axons was performed for 45 axonal segments. For 10 axon ROIs, motion artifacts were detected and corrected manually using custom-written LabVIEW 2018 software. All calcium traces were resampled at 100 Hz and then low-pass filtered at 1 Hz. Example traces are presented as ΔF/F (where F is the low passed fluorescence signal the 15 percentiles on the entire signal). However, in all other analyses, Flurecrnt_norm was used (calculated as (F-Fmin)/(Fmax-Fmin)) to not over- or under-weight highly active calcium traces[51]. For the control imaging experiment of CaMKII cells, suite2p was used to detect and extract calcium signals[85].

**Calcium imaging analysis**. Cross-correlation between whisking signal and axonal florescent was calculated after low-pass filtering the calcium traces at 1 Hz and subtracting the mean from each trace.

The second method for evaluating the callosal florescent activity during whisking/quiet wakefulness was performed by setting a threshold of four times the median absolute deviation (mad) over the signal median. This approach allowed us to remove baseline activity altogether and examine only high-amplitude florescent activity during whisking/quiet wakefulness.

**Optogenetic stimulation**. For optogenetic stimulation, we used an analog modulated blue DPSS laser (λ = 473 nm, Shanghai Dream Lasers Technology) coupled to a multimode fiber (Ø400 μm core, 0.39 NA, FT400UMT, ThorLabs, Newton, NJ). The intensity of the light was ~7 mW at the tip of the fiber. For local cortical stimulation, a continuous light pulse 2 seconds long was given through the fiber that was placed <2 mm from the surface of the barrel cortex (1.3 C, 3.3 L from bregma).

**Histology**. At the end of the experiment, mice were over-anesthetized (pentobarbital 1 gr/kg, I.P) and perfused transcardially with 2.5% paraformaldehyde. The brain was removed and postfixed for 24 h in the perfusion solution. Brains were then immersed in PBS solution with additional 30% sucrose for 24 h and then cut in a freezing microtome (80 μm thick, SM 2000R; Leica, Heidelberg, Germany).

Brain slices were mounted on slides and scanned using ZEN software (Zeiss) by a fluorescent microscope (VS120-S6, Olympus). Images were exported and processed using ImageJ software.

**Statistical analysis**. The data were evaluated using a Wilcoxon signed-rank test (WSRT), or paired *t*-test, individually tested for normal distribution (Shapiro–Wilk

test). Mann–Whitney U test was used for unpaired data and two-way ANOVA multiple comparisons.

Modulation index for quiet wakefulness vs. whisking epochs was calculated as follows:

$$MI_{state} = \frac{\left| Peak\ of\ normalized\ STA_{whisk} \right| - \left| Peak\ of\ normalized\ STA_{quiet} \right|}{\left| Peak\ of\ normalized\ STA_{whisk} \right| + \left| Peak\ of\ normalized\ STA_{quiet} \right|} \qquad (2)$$

Modulation index for before vs. after CNO application was calculated for both quiet wakefulness and whisking epochs as follows:

$$MI_{CNO} = \frac{\left| Peak\ of\ normalized\ STA_{beforeCNO} \right| - \left| Peak\ of\ normalized\ STA_{afterCNO} \right|}{\left| Peak\ of\ normalized\ STA_{beforeCNO} \right| + \left| Peak\ of\ normalized\ STA_{afterCNO} \right|} \qquad (3)$$

**Reporting summary**. Further information on research design is available in the Nature Research Reporting Summary linked to this article.

## Data availability
Source data are available in the Source Data section. All processed data that are presented in this study have been deposited in an OSF database [https://osf.io/4gj7p/]. Raw data are available from the corresponding author upon reasonable request.

## Code availability
The routines and code used for analysis are available from the corresponding author upon reasonable request.

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

## Acknowledgements

I.L. is the incumbent of the Norman and Helen Asher Professorial Chair at the Weizmann Institute of Science. Y.K. is incumbent of the Marianne Manoville Beck Research Fellow Chair in Brain Research. We thank A. Parabucki and for making the schematic illustrations and both her and A. Marmelshtein for constructive comments on an earlier version of this manuscript. This research was supported by DFG (SFB 1089), EraNet (DeCipher Neuron 01EW1606), Human Frontier Science Program Grant, Israel Science Foundation (ISF 1539/17), BSF grant 2019251, Minerva, and the Marianne Manoville Beck Laboratory for Research in Neurobiology in Honor of her Parents Elisabeth and Miksa Manoville, all awarded to I.L.

## Author contributions

I.L, Y.O., and Y.K designed the experiments. Y.K., Y.O., and K.C.K.M. carried out the electrophysiology experiments. Y.O. and M.S. carried out the calcium imaging experiments. Y.O. and Y.K. carried out the lesion experiments. Y.O. performed the histology. I.L., Y.O., and Y.K. carried out the analyses. I.L., Y.O., M.S., and Y.K. wrote the manuscript.

## Competing interests

The authors declare no competing interests.
