## [Peer Review File · Nature Communications]

Reviewer #1 (Remarks to the Author):

The paper by Oran and colleagues is focused on the correlation of neuronal activity between the two cortical hemispheres. Using different electrophysiological and imaging methods they study the activity in the primary somatosensory cortex (specifically, barrel cortex) of mice, obtaining simultaneous recording from both hemispheres. The degree of correlation between the hemispheres is shown to strongly depend on the behavior of the mouse, specifically whether it is whisking or in quiet wakefulness. Using optogenetics and chemogenetics they further show that the high correlation observed during quiescence is mediated by the corpus callosum. During whisking, the activity of callosal cells and the degree of correlation are reduced, despite an overall increase in the activity rate in both hemispheres. The paper is interesting and timely, based on high quality experimental data and analysis. I have a few comments below, which should be addressed prior to publication.

The paper convincingly shows the role of the corpus callosum in mediating cross hemispheric correlations, however, it is not clear what the alternative mechanisms and pathways are and how their involvement could be assessed in different brain-states. During whisking, when corpus callosum is “quiet”, despite the increased overall bilateral activity, where does the slow signal for increased activity arrive to both hemispheres? Is it mediated by bilateral thalamic or brainstem projections? Bilateral neuromodulation? Is it due to a feedback from the “motor whisker cortex”?

The pyramidal cells that project to the other cortical hemisphere were studied in rodents in several papers such (Brown & Hestrin, LeBeouvier...Markram et al 2003, Lee...Sohal et al 2014, to name just a few). These papers describe their properties as well as connectivity patterns within the cortical circuitry. How do the authors explain the reduced activity of this specific population, despite an increase in the overall activity rate? Is there any data or studies that support active suppression of the callosal population during whisking? There should be a discussion about the most likely mechanisms underlying the opposed modulation between cortico-callosal cells and other pyramidal subpopulations in the barrel cortex.

One experiment that could shed light on this opposite modulation is by imaging cortico-callosal cells via retrograde AAV-cre expression in the other hemisphere and compare those results to similar imaging of cortico-spinal or cortico-brainstem ipsilaterally projecting pyramidal cells. According to the authors' predictions, the former population will show decreased activity during whisking while the latter will increase. One potential hurdle in this experiment is the depth of the respective pyramidal cells (layers 5a and 5b), which may prove difficult to image with sufficient quality.

Figures 1d and 1e, the difference between quiet and whisking STA seems much larger within hemisphere than between. It is not clear what is presented in 1f. Does it refer to modulation of firing rates? Correlations? Within or between hemispheres? Should be better explained.

In figure 4, the photo-inhibition of callosal terminals is shown as a population average for whisking and quiet epochs. The effect is not so clear in the average, could the authors add a clearer example? In particular, the inhibition lasts for ~2 seconds but only 200 ms are analyzed. What is the authors' interpretation of these results? Also, in this figure, the division is not to “quiescence vs. whisking” but rather to “high vs low” whisking. Is this intentional? If so, what is the meaning of this difference? Are the three examples in 4b given from whisking or non-whisking epochs? If in quiescence, wouldn't we expect some impact on the contralateral cell? Lastly, the photostimulation in 4b and 4c are not of the same duration.

The use of CNO injected in the contralateral hemisphere showed state-dependent effects on the interhemispheric correlation. However, the authors do not report what was the effect of CNO on the firing rates in both hemispheres. Was there an attempt to assess the overall effect of CNO in the virally transduced hemisphere? Some more details should be presented regarding the chemogenetic experiments

MINOR

- "optoinhibition" is not a common word, especially for abstract. Should perhaps use "optogenetic inhibition..." or "optogenetic silencing"
- Line 180: The analysis separates the >1Hz filtered signal. Why not do several band-pass filtering and compare CC values for different bands? As done by Lampl in 1999 Neuron paper?
- One other target of corpus callosum is the corticostriatal pathway to the ipsi- and contralateral striatum, which is very different between motor and somatosensory cortices (See Reig & Silberberg, Cerebral Cortex 2016). This could suggest an active role of striatum in shaping interhemispheric correlation during whisking and quiet epochs.
- Fig 1gi, there appears a diagonal white line. What does it designate? Is it just an image mistake?
- Supp figure 4 title incomplete
- Typos/grammar should be checked thoroughly, here are some examples:
- Line 50: "another hallmark... [is]..."
- Line 67: "abundant research [was] done"
- Line 111: "to [observe] pair-wise correlation..."
- Line 248: "according [to] the whisking activity"
- Line 273: "...if such [an] effect is..." or "...if this effect is..."
- Line 378: "which [were] observed in..."
- Line 401 "was suggested to [result] from..."
- Line 416: "before each...cortical [event]..."

Reviewer #2 (Remarks to the Author):

This manuscript is yet another attempt to unlock the mystery of the corpus callosum function, a research endeavor with a checkered history. Armed with an impressive battery of state-of-the art complementary techniques the authors convincingly demonstrate that the corpus callosum seem to have modulatory influence only during quite waking when mouse whiskers were not moving. The authors should be lauded for their efforts to overcome the 'common-input' vs. direct modulation issue and their efforts not to rely only on injury-based manipulation that is unfortunately quite prevalent in the corpus-callosum studies. I have found that the application and the analysis of the techniques were described at a satisfactory level. Most of my feedback is on general/conceptual and control issues.

There are several issues that the authors should respond to:

The authors seem to ignore some convincing rodent literature that demonstrates that when freely moving rodents 'whisk' in the air, there is barely any evoked activity (sub- and suprathreshold) in sensory cortex, including barrel cortex (examples: Ferezou and Petersen Neuron 2006, Faselow and Nicolelis J. Neurosci. 1999). There seem to be, therefore, a significant difference between freely moving rodents and head-fixed ones (the current manuscript) where in the latter case the authors

convincingly demonstrate that whisking in the air leads to a significant increase in evoked neuronal activity. These contradictory findings therefore bag the question on whether results obtained from head-fixed rodents, a preparation that has increases in popularity in many labs around the world, are valid also as a model of the awake brain on the move. Such results may be only relevant for the head-fixed preparation, which unfortunately is not even close to any natural model.

A similar issue has to do with the lack of behavioral outcome following what seems to be a convincing case of shutting down activity in cortex (lines 245-246). The authors report that the inhibition had no effect on whisking activity. However, Carl Petersen (Science 2010) has convincingly demonstrated that barrel cortex itself is actively participating in the whisking movement and therefore one would expect that inhibition of barrel cortex should have an effect on whisking. One potential explanation for what seems to be contradictory findings is that the authors of the current manuscript focused mostly on the upper layers of barrel cortex and potentially the lower layers continued to work somehow and do what they are supposed to do. This leads me to ask the authors to better highlight that their results seem to be only related to the upper layers and to explain the potential ramifications of such focused study.

The authors assume that corpus callosum is connecting the entire area of the barrel cortices, is there any clear evidence for such mapping? In the visual cortex of rodents, cats and monkeys, corpus callosum connect only small parts of the visual cortex, parts that have their receptive fields along the vertical meridian. If corpus callosum also connects only specific parts of the somatosensory cortices in mice, the exact site of recordings/inactivation become much more important and can strongly influence the results.

In a related issue, roughly how much cortical volume was inhibited with light and CNO?

Fig. 4b: the traces from both hemispheres before and after inactivation seem to be different from each other. Also, I was surprised for what seems like a complete lack for a post-inhibitory 'rebound' that typifies optogenetic-based inhibition.

Lines 479-484: is there any evidence for the authors suggestion on the relationship between corpus callosum modulation during quiet epochs and homeostasis?

Minor:

Fig 1: the letter 'b' is missing for the second panel.

Fig.2: The cross-correlations are too small to infer where exactly the peak is and its relationship to zero time-lag.

Reviewer #3 (Remarks to the Author):

The study presented in the manuscript # NCOMMS-20-38247-T by Oran, Katz and collaborators, investigated the implication of transcallosal projections in interhemispheric correlation of neuronal activity and its modulation by the behavioral state, in awake head-fixed mice.

Focusing their attention on the whiskers representation within the primary somatosensory cortex (S1), they used complementary experimental approaches: single-unit, whole-cell and local field potential bilateral recordings, in addition to opto and pharmacogenetic approaches. From this substantial set of data, they demonstrate a decrease of interhemispheric correlation of neuronal activity at both supra and sub-threshold levels upon whisking. They further show results indicating that this decrease of interhemispheric correlation is likely to be due to a reduced recruitment of transcallosal projections in this behavioral state.

These two observations are novel and as such of interest for the community. Indeed, on the one hand, a strong synchronization of spontaneous waves of activity between hemispheres through transcallosal projections during the awake “quiet” state has been reported before by means of mesoscale imaging experiments (Mohajerani et al., 2010). On the other hand, a transition from “quiet” to “whisking” behavior has been shown to be associated with a disappearance of these slow waves, together with membrane potential depolarization and local neuronal desynchronization (Crochet & Petersen, 2006, Poulet & Petersen, 2008, Poulet et al., 2012; Eggermann et al., 2014). Given such local desynchronization and depolarization during whisking, which has been shown to emerge from an increase of both thalamic and cholinergic drive, one could have expected that measuring long-range interhemispheric correlation would be made more difficult, but not necessarily that whisking would be associated with a decrease of activity in transcallosal axons. The study reported here therefore provide data of interest and is globally technically sound, however, some control experiments are needed to support the main conclusions and the quality of the manuscript has to be improved. Here a list of concerns that deserved to be addressed.

- Mice are usually, after a few sessions of habituation to head fixation, rarely whisking spontaneously, and could even present episodes of sleep (Fernandez et al., 2017). In the present study however, the authors do not mention any habituation of the animals to the head fixation before the recordings. This might explain why they assume that the mice are never falling asleep during the recordings, splitting the data in only two states: quiet awake and whisking. This might also be why they could get relatively frequent episodes of whisking during the recordings. However, the proportion of time spend in quiet vs whisking behavior should be clearly stated for all the figures. In addition, in case any special experimental means have been used to trigger whisking episodes during the recordings, this should be clearly mentioned in the methods section. Given this experimental design, it is likely that episodes of whisking are correlated with global body movements of the mouse trying to escape from this unusual head fixation condition. To my point of view, the expression “exploratory” is not really appropriate to qualify this behavior.

- The figure 6 and movie 1 illustrate 2-photon calcium imaging of transcallosal axons in awake mice. These data are supporting the main conclusion of the manuscript: a decrease of transcallosal activity during whisking. As acknowledged by the authors, and for the reasons mentioned above, the whisking episodes reported here are likely to be accompanied by strong movements that might impact the 2-photon recording of calcium transients in thin axonal portions. Performing similar experiments, with a viral construct driving the expression of GFP instead of GCaMP6s, would allow assessing how these movements affect the axonal imaging during whisking and if they could ultimately lead to a misinterpretation of the results.

The methods section indicate that pupil diameter has been monitored together with whisking activity during these imaging experiments (line 674), and that the mice were held on a treadmill (lines 672 & 676). Either these mentions have been made by mistake (text inherited from a previous unrelated work), or, if it is really the case, they deserve to be more clearly stated and commented in the main text and complementary analyses should be provided. From what I have understood, there was no treadmill for all the other experiments of the study, which means that the behavioral “whisking” state is likely to be different in this last set of experiments, being most probably associated with running.

In addition, there are many details in this figure that should be clarified. Lines 685-687 of the methods section, it is indicated that a different preprocessing has been applied to the “illustrated examples” than “in all other analyses”, why? The two examples illustrated in panel e have been

recorded simultaneously. Why not illustrating the activity of the two axonal portions throughout the same episodes of whisking then? Given the large variability of the population data, illustrating examples from different mice would be very welcome.

The panel a shows a strange assembly of images, are they all coming from the same histological section? Please specify. The movie is lacking any temporal or spatial scale indication.

- The results in figure 4 are interesting however control experiments assessing the effect of light in the absence of ChR2 expression would be very useful. It is also crucial to provide a proper analysis of the behavior to check that the photostimulation is not triggering any whisking when delivered in the quiet state. The extended fig. 4 in its actual form is non informative, please specify what are the plots corresponding to, and provide raw data measurement of whisking activity during the photostimulation. Is the quantification in panel d corresponding to a mean Vm in the first 200ms following light or a negative peak Vm value? Which time window has been used for "baseline" measurement? Why not showing the whisking activity in panel b?

- In the figure 5, panel b and d, it should be specified if these are example STAs for given units or population averages. If examples, then it should be indicated to which line they correspond in the next panel. One could expect that the CNO application should not affect the "within hemisphere" STAs as opposed to the inter-hemisphere ones. It would be nevertheless interesting to show it.

- Minor points:

The text of the manuscript, and legends in particular must be carefully reread.

The figure 2 and 3 should be merged together.

The authors could refer to an interesting recent related study: Ferrier J, Tiran E, Deffieux T, Tanter M, Lenkei Z. Functional imaging evidence for task-induced deactivation and disconnection of a major default mode network hub in the mouse brain. Proc Natl Acad Sci U S A. 2020 Jun 30;117(26):15270-15280.

Some key aspects of the experiments are missing in the methods section:

-LFP recordings: how are the LFP bundles inserted in the craniotomies, at which depth?

-Whisking detection via camera: the threshold algorithm seems to apply to a 1D "signal", how is this signal related to the acquired 2D images?

-The lesion experiments should be described with more detail in a dedicated section.

The extended data file has to be corrected. The extended fig. 3 is not supporting the assertion of the main text (lines 149-151) where it is mentioned. The extended fig. 4 is missing a proper title and legend etc...

Dear Editor and Reviewers

We would like to thank the reviewers for the constructive comments. We hope that our point-by-point answers and the additional experiments and data analysis addressed the raised concerns. Before presenting the point-by-point answers, we want to note that figure numbers were changed due to merging figures 2 and 3 as suggested by the reviewer. Also, the current version contains 13 supplementary figures.

REVIEWER COMMENTS

Reviewer #1 (Remarks to the Author):

The paper by Oran and colleagues is focused on the correlation of neuronal activity between the two cortical hemispheres. Using different electrophysiological and imaging methods they study the activity in the primary somatosensory cortex (specifically, barrel cortex) of mice, obtaining simultaneous recording from both hemispheres. The degree of correlation between the hemispheres is shown to strongly depend on the behavior of the mouse, specifically whether it is whisking or in quiet wakefulness. Using optogenetics and chemogenetics they further show that the high correlation observed during quiescence is mediated by the corpus callosum. During whisking, the activity of callosal cells and the degree of correlation are reduced, despite an overall increase in the activity rate in both hemispheres. The paper is interesting and timely, based on high quality experimental data and analysis. I have a few comments below, which should be addressed prior to publication.

- The paper convincingly shows the role of the corpus callosum in mediating cross hemispheric correlations, however, it is not clear what the alternative mechanisms and pathways are and how their involvement could be assessed in different brain-states. During whisking, when corpus callosum is “quiet”, despite the increased overall bilateral activity, where does the slow signal for increased activity arrive to both hemispheres? Is it mediated by bilateral thalamic or brainstem projections? Bilateral neuromodulation? Is it due to a feedback from the “motor whisker cortex”?

Indeed, several mechanisms can lead to increased bilateral activity in the somatosensory cortex during whisking. Although this is beyond the scope of this work, we further discuss the possibilities raised by the reviewer, which were briefly mentioned in our original submission:

“ We hypothesize that the elevated cortical activity during whisking may reflect different processes, including: (1) increased firing rate of thalamic cells^{1,2}, (2) activation of different neuromodulatory axons, such as cholinergic and noradrenergic^{3,4}, which leads to increase of cortical firing⁵⁻⁸, and (3) inputs from the motor cortex which modulate firing in S1, as was

previously suggested^{9,10}. The differences we found between the different recording methods within our own study, however, may reflect a possible bias for a specific cell population in each given method. Nevertheless, our results indicate that whisking suppresses the firing of callosally projecting neurons.” (Discussion line 414).

- The pyramidal cells that project to the other cortical hemisphere were studied in rodents in several papers such (Brown & Hestrin , Le Be...Markram et al 2003, Lee...Sohal et al 2014, to name just a few). These papers describe their properties as well as connectivity patterns within the cortical circuitry. How do the authors explain the reduced activity of this specific population, despite an increase in the overall activity rate? Is there any data or studies that support active suppression of the callosal population during whisking? There should be a discussion about the most likely mechanisms underlying the opposed modulation between cortico-callosal cells and other pyramidal subpopulations in the barrel cortex.

We would like to thank the reviewer for this important comment. As far as we know, no other studies explored the activity of callosal projecting population during whisking and non-whisking epochs and studied the underlying mechanisms for this modulation. However, the papers mentioned by the reviewer potentially also explain some of our results.

Several studies described the modulation of local firing during whisking. de Kock and Sakmann (2009) showed that a subset of neurons increases their average firing rate upon free whisking in head-restrained rats. These neurons were preferentially located in layer 5A. Another study from Yu and Svoboda¹¹ has shown that pyramidal cells (excitatory) can be excited or inhibited by whisking throughout the cortical layers. VPM neurons were strongly excited by whisking, while FS neurons show diverse effects, depending on the cortical layer. These results suggest that there might be subpopulations within the pyramidal cells that are activated differently during whisking. One possibility is that an internal network of inhibitory interneurons decreases the firing rate of callosal cells.

The studies that the reviewer raised partially explain differential modulation of callosally projecting cells. Le Be et al. (2007) show that cortico-callosally projecting cells are less excitable and exhibit lower interconnectivity when compared to other long-range projecting pyramidal cells. Moreover, Lee et al. (2014) discovered also that stimulation of callosal inputs onto corticocallosally projecting cells depress more and evoke less firing when compared to subcortically projecting pyramidal cells. We summarized these finding in a short sentence added in the discussion:

“The mechanisms underlying the reduction of callosal activity during whisking are yet unknown and can reflect the activity of inhibitory inputs. Since whisking has diverse effects on the firing rate of excitatory and inhibitory cells in the barrel cortex², callosally-projecting neurons could be modulated in a specific manner. Indeed, when compared to other long-range projecting pyramidal

cells, callosally-projecting cortical neurons are intrinsically less excitable and exhibit a weaker response to contralateral callosal inputs^{12,13}.”

- One experiment that could shed light on this opposite modulation is by imaging cortico-callosal cells via retrograde AAV-cre expression in the other hemisphere and compare those results to similar imaging of cortico-spinal or cortico-brainstem ipsilaterally projecting pyramidal cells. According to the authors’ predictions, the former population will show decreased activity during whisking while the latter will increase. One potential hurdle in this experiment is the depth of the respective pyramidal cells (layers 5a and 5b), which may prove difficult to image with sufficient quality.

This is an excellent suggestion that we already tried while working on this project. Unfortunately, and as was predicted by the reviewer, we obtained weak signals and very sparse expression following retrograde expression of genetically encoded calcium indicator. This is one of the reasons why we used the axonal imaging.

That being said, we performed new imaging experiments of local CaMKII cells using GCaMP6s from upper cortical layers in awake head-fixed mice. This was also done in response to the second reviewer's comments regarding our observation using extracellular recordings demonstrating that the mean firing rate of neurons increased during whisking. Our results (shown in this rebuttal letter and in Supplementary Figure 13) indicate that the calcium activity of CaMKII cells increased during whisking, although more moderately compared to the extracellular data.

- Figures 1d and 1e, the difference between quiet and whisking STA seems much larger within hemisphere than between. It is not clear what is presented in 1f. Does it refer to modulation of firing rates? Correlations? Within or between hemispheres? Should be better explained.

We apologize for not explaining these figures well enough. Briefly, these figures describe the mean STAs when spikes in one side were used to compute the firing histogram of neurons recorded from the other side (or from the within hemisphere for local STA). Indeed, the difference between quiet and whisking STAs between hemispheres is smaller than their respective STAs within each hemisphere. Figure 1f corresponds to the modulation of STA from all combinations of unit pairs between hemispheres when sorted based on the magnitude of the STAs (summed for quiet and whisking STAs). The modulation index is calculated as the difference between peak STA activity during whisking epochs and quiet wakefulness divided by their sum (dashed line marks mean), calculated for all combinations of unit pairs between hemispheres. The text and legend describing this figure were changed in the revised manuscript to explain the figure better.

- In figure 4, the photo-inhibition of callosal terminals is shown as a population average for whisking and quiet epochs. The effect is not so clear in the average, could the authors add a clearer example?

We would like to thank the reviewer for drawing our attention to this figure. As a result, we changed the figure to better illustrate the effect we recorded (changes were also made in response to the other reviewers' comments).

Figure 4 shows how inactivation of the barrel cortex of one hemisphere using optogenetic-inhibition in GAD-ChR2 mice affects the membrane potential of cells recorded in the other hemisphere. This allowed us to test how unilateral silencing affects the activity in the contralateral hemisphere. Thus, it was used first to probe if the two hemispheres are coupled and then find if the coupling depends on the state of the animal. To clarify, the revised figure (panel 4c) shows a single cell example that demonstrates the transient light-evoked hyperpolarization we found. The aim in panel B, using paired intracellular recordings, was to show that silencing was efficient in the illuminated side. Note that the effect on the other side is barely observed in single trials, implying in general that the two somatosensory cortices are weakly coupled. However, we thought we might have missed something at this point, as our recordings from both sides show state-dependent interhemispheric correlations (Figure 1). Therefore, we decided to look deeper into the data, speculating that coupling should be observed during non-whisking epochs. Indeed, we observed a transient hyperpolarization in the first 200 ms locked to the light only during the quiet state, supporting our conjuncture that the two hemispheres are more coupled during this state. Hence, that coupling was higher in this state is in agreement with the higher correlation we found using the paired extracellular and intracellular recordings.

- In particular, the inhibition lasts for ~2 seconds but only 200 ms are analyzed. What is the authors' interpretation of these results?

Short hyperpolarization following light inactivation of one hemisphere was recently reported in a study of the auditory cortex¹⁴, demonstrating a diverse effect on spontaneous activity in A1. Furthermore, previous studies established that callosal inputs make direct excitatory connections onto cortical pyramidal cells^{15,16} and drive disynaptic feedforward inhibition onto local GABAergic interneurons^{17,18}. Therefore, the most reasonable explanation for this transient hyperpolarization is a withdrawal of direct excitatory callosal input onto specific cells, which is probably quickly compensated by local network effects. This issue is also further elaborated in the discussion, where we described these results.

- Also, in this figure, the division is not to “quiescence vs. whisking” but rather to “high vs low” whisking. Is this intentional? If so, what is the meaning of this difference?

In all other figures in the paper, whisking refers to epochs of whisking, while quiet wakefulness refers to epochs where the animal did not whisk at all. In the experiment described in figure 4 (now Figure 3, after merging 2 and 3), light stimulation was randomly given according to protocol settings (4 seconds trial, optogenetic inhibition from 1-3 sec.). Since light onset was not locked to the animals' state, whisking could start before or during light stimulation. Therefore, to evaluate the light's effect, we grouped the trials according to the amount of whisking (sorted by the overall STD of whisking signal in each trial) and compared the light's impact between the high Vs. low whisking trials. Hence high whisking trials consist of trials during which the animal spent a significant amount of time whisking whereas in low whisking trials the animal spent much less time whisking. We also clarified this issue in the methods section – “Whisking detection via reflective sensor”:

“In figure 3, trials were sorted according to the standard deviation of whisking level during the trial (start to 1.5s). Hence high-whisking trials consist of trials during which the animal spent a

significant amount of time whisking, while in low-whisking trials the animal spent little to no time whisking.”

- Are the three examples in 4b given from whisking or non-whisking epochs? If in quiescence, wouldn't we expect some impact on the contralateral cell?

Panel b of Figure 3 (previously 4) shows that light leads to robust inactivation of local activity in the illuminated side. This experiment was conducted only in one intracellular paired recording (this method was verified before in our lab ¹⁹) and no separation into high whisking vs. low whisking epochs was made. The other panels at this figure show intracellular recordings made in the contralateral hemisphere to the one that was inactivated. The lack of noticeable effect on the contralateral recorded cell seen in single trials in panel b is consistent with our findings that a transient effect was only revealed when we sorted the trials based on the amount of whisking. Hence, as shown in the example that is demonstrated in the new Figure 3 panel c of the revised manuscript, one can observe the impact of the optogenetic inhibition only after dividing the data to high vs. low whisking. Additionally, due to the way we divided the trials to high vs. low whisking, whisking and quiet wakefulness epochs can be found in both groups.

In addition, as requested by the third reviewer, whisking signal was added to individual trials in panel B. Additional experiments and analysis of the effect of light on whisking initiation, both in WT and ChR2-GAD mice are shown below as a response to the two other reviewers.

- Lastly, the photostimulation in 4b and 4c are not of the same duration.

We thank the reviewer for noticing this issue. Though these are different experiments (panel b-paired intracellular, panel c-single intracellular), the optogenetic inhibition duration was similar in both, and we had the wrong scale bar at panel b of old Figure 4. We fixed this in the revised manuscript for the new Figure 3.

- The use of CNO injected in the contralateral hemisphere showed state-dependent effects on the interhemispheric correlation. However, the authors do not report what was the effect of CNO on the firing rates in both hemispheres. Was there an attempt to assess the overall effect of CNO in the virally transduced hemisphere? Some more details should be presented regarding the chemogenetic experiments

Thanks for raising this important question. We checked the effect of CNO on the firing rate within the hemisphere in which we applied it (right hemisphere), as well as its effect in the hemisphere in which the virus was injected (left hemisphere, injected with the hM4D virus). Figure R1 shows these results (added as Supplementary Figure 9 to which we referred in the results). We found that the application of CNO did not have a prominent effect on the mean population firing rate in the right or left hemispheres when testing it during each of the two states (in all panels no significant effect was found, except for panel e in which a small but significant effect was measured, $p = 0.022$).

Figure R1. The effect of CNO on the average firing rate in both hemispheres was small. **a**. The mean firing rate before and after applying CNO in the right hemisphere (contralateral to CNO application). **d**. In the left hemisphere (contralateral to CNO application). **b,c,e,f** Bar plot of the firing rates as shown in a and d. Note that in a and d, axes were truncated and thus a few data points are not shown, but are presented in the bar plots ($p > 0.05$ for b,c,f, $p = 0.02$ for e).

MINOR

- “optoinhibition” is not a common word, especially for abstract. Should perhaps use “optogenetic inhibition...” or “optogenetic silencing”

We replaced the term optoinhibition with optogenetic inhibition.

- Line 180: The analysis separates the $>1\text{Hz}$ filtered signal. Why not do several band-pass filtering and compare CC values for different bands? As done by Lampl in 1999 Neuron paper?

Thanks for bringing this issue to our attention. The analysis presented in Lampl et al. 1999 was not done in an efficient way, and the same conclusions could be made using Coherence analysis²⁰. We used coherence rather than correlation since it reduces filter bias (due to narrow bandpass filtering) and gives a similar result²¹.

Although we checked the coherence for our paired intracellular recordings, we decided not to add the results into the main figures so to not to overload the manuscript. However, we now added

the coherence analysis results to the manuscript as Supplemental Figure 6. This analysis shows that the coherence in V_m between the two hemispheres is higher than for shuffled data up to 7 Hz (as quantified by a rank-sum test) and decreases as frequency increases.

Figure R2. Interhemispheric coherence was significant up to 7 Hz. The coherence between the membrane voltages of simultaneously recorded pairs (green trace) was significantly different from shuffled data (purple trace) up to 7 Hz (WSRT. Shaded area = SEM). Shuffled-data coherence (purple line) was produced by calculating the coherence between V_m from one cell at time zero versus the V_m of the other cell at a random time.

- One other target of corpus callosum is the corticostriatal pathway to the ipsi- and contralateral striatum, which is very different between motor and somatosensory cortices (See Reig & Silberberg, Cerebral Cortex 2016). This could suggest an active role of striatum in shaping interhemispheric correlation during whisking and quiet epochs.

We thank the reviewer for raising this possibility, and therefore we added this sentence into the discussion:

“Finally, these correlations can also be shaped by other non-cortical brain regions, such as the striatum which exhibits bilateral vibrissa responses²² and thus may play an important role in the state-dependent interhemispheric correlations. “

- Fig 1gi, there appears a diagonal white line. What does it designate? Is it just an image mistake?

Image mistake, fixed in the revised manuscript

- Supp figure 4 title incomplete – *fixed*.

- Typos/grammar should be checked thoroughly, here are some examples:

- Line 50: “another hallmark... [is]...” : *fixed*
- Line 67: “abundant research [was] done” : *fixed*
- Line 111: “to [observe] pair-wise correlation...” : *fixed*
- Line 248: “according [to] the whisking activity” : *fixed*
- Line 273: “...if such [an] effect is...” or “...if this effect is...” : *fixed*
- Line 378: “which [were] observed in...” : *fixed*
- Line 401 “was suggested to [result] from...” : *fixed*
- Line 416: “before each...cortical [event]...” : *fixed*

Reviewer #2 (Remarks to the Author):

This manuscript is yet another attempt to unlock the mystery of the corpus callosum function, a research endeavor with a checkered history. Armed with an impressive battery of state-of-the-art complementary techniques the authors convincingly demonstrate that the corpus callosum seem to have modulatory influence only during quite waking when mouse whiskers were not moving. The authors should be lauded for their efforts to overcome the ‘common-input’ vs. direct modulation issue and their efforts not to rely only on injury-based manipulation that is unfortunately quite prevalent in the corpus-callosum studies. I have found that the application and the analysis of the techniques were described at a satisfactory level. Most of my feedback is on general/conceptual and control issues.

There are several issues that the authors should respond to:

- The authors seem to ignore some convincing rodent literature that demonstrates that when freely moving rodents ‘whisk’ in the air, there is barely any evoked activity (sub- and suprathreshold) in sensory cortex, including barrel cortex (examples: Ferezou and Petersen Neuron 2006, Fanselow and Nicolelis J. Neurosci. 1999). There seem to be, therefore, a significant difference between freely moving rodents and head-fixed ones (the current manuscript) where in the latter case the authors convincingly demonstrate that whisking in the air leads to a significant increase in evoked neuronal activity. These contradictory findings therefore bag the question on whether results obtained from head-fixed rodents, a preparation that has increases in popularity in many labs around the world, are valid also as a model of the awake brain on the move. Such results may be only relevant for the head-fixed preparation, which unfortunately is not even close to any natural model.

We would like to thank the reviewer for this comment. First, we want to report that in response to the first reviewer, we made additional experiments in which we used 2P imaging of CaMKII-GCamp6s cells in upper cortical layers and found moderate (but significant) elevation of calcium signals during whisking. These effects were found by computing the cross-correlation between the calcium and whisking signals and by taking the mean signal after thresholding (similar analysis as was made for the callosal axons, in Figure 5, old Figure 6). These data are presented in Figure R3 and were added as a new Supplementary Figure (#13) and refer to this figure in the Results. It is not clear why the imaging and extracellular recordings do not show the same degree of elevation in activity during whisking; however, each method may contain some bias for the type of cells recorded. That being said, both types of recordings clearly indicate that the reduction in callosal activity during whisking is unique to this population of signals.

Indeed, a few studies have found that there is no increase in firing rates during whisking which was also reported for head-fixed mice. However, we would like to emphasize that other studies did report increased firing rate in S1 during whisking (Curtis and Kleinfeld, 2009; de Kock and Sakmann, 2009). We assume that specific experimental parameters may explain these differences. These include freely moving vs. head-fixed preparations, differences in habituation procedures, and differences in the location and type of neurons within the barrel cortex. To refer to this discrepancy, we have changed our original discussion sentence on this matter (L376) accordingly and included the sentence that we wrote in response to the first reviewer that asked about the possible origins for the elevated signal during whisking:

“Whether or not the mean firing rate of cortical neurons increases during free whisking is controversial. While some studies, both in freely moving and head-fixed animals, found little change in the population firing rate^{2,11,25–27}, others showed elevated firing^{23,24}. Such discrepancies might be explained by differences in experimental parameters, such as recordings from freely moving versus head-fixed animals, habituation procedures, as well as differences in the cell types and cortical layer of the recorded neuronal population. While our extracellular recordings, demonstrating higher firing rate during whisking (Fig. 1c), are more consistent with the latter group of studies, our imaging data (Supplementary Fig. 13) suggests a more moderate elevation in neuronal activity during whisking. We hypothesize that the elevated cortical activity during whisking may reflect different processes, including: (1) increased firing rate of thalamic cells^{1,2}, (2) activation of different neuromodulatory axons, such as cholinergic and noradrenergic^{3,4}, which leads to increase of cortical firing^{5–8}, and (3) inputs from the motor cortex which modulate firing in S1, as was previously suggested^{9,10}. The differences we found between the different recording methods within our own study, however, may reflect a possible bias for a specific cell population in each given method. Nevertheless, our results indicate that whisking suppresses the firing of callosally projecting neurons.”

Figure R3: Calcium imaging of CaMKII cells in upper cortical layers of awake mice shows that neuronal activity is correlated with whisking. **a.** A picture of the imaged cortex. **b.** Representative calcium signals from 8 cells and the whisking signal below. **c.** The population average of the cross-correlations between whisking and calcium signals (n = 2 mice, n = 229 cells). **d.** Mean calcium signals that crossed a threshold (4 times the median absolute deviation) for quiet and whisking epochs (n = 229 cells).

We agree that the results we obtained in head-fixed mice are limited and thus cannot be generalized to other conditions, such as freely moving animals. Yet, head-fixed awake preparations allow for greater control and monitoring of animal state and also allows better conditions for electrophysiological recordings and imaging. We would like to emphasize that in the last Barrels meeting (2020), we saw several studies in which head-fixed mice were studied, suggesting that this model is still a standard model. Due to this important comment, we added a sentence in the discussion in which we wrote:

“Whether or not state-dependent callosal activity and its contribution to interhemispheric correlations are also present in freely-moving animals remains to be explored in future studies”

- A similar issue has to do with the lack of behavioral outcome following what seems to be a convincing case of shutting down activity in cortex (lines 245 246). The authors report that the inhibition had no effect on whisking activity. However, Carl Petersen (Science 2010) has convincingly demonstrated that barrel cortex itself is actively participating in the whisking movement and therefore one would expect that inhibition of barrel cortex should have an effect on whisking.
- One potential explanation for what seems to be contradictory findings is that the authors of the current manuscript focused mostly on the upper layers of barrel cortex and potentially the lower layers continued to work somehow and do what they are supposed

to do. This leads me to ask the authors to better highlight that their results seem to be only related to the upper layers and to explain the potential ramifications of such focused study.

These concerns were carefully addressed using a set of additional experiments and analyses. We showed before¹⁹ that the light in these transgenic mice silenced the firing of cells across all layers, and therefore we do not think that we inactivated only upper cortical layers. This was mentioned in the original submission: “Using the same line of mice, we previously showed that such light stimulation abolished firing across all cortical layers¹⁹”.

That being said, we would like to thank the reviewer for asking about the effect of cortical silencing on whisking initiation, as was demonstrated in the studies of Carl Petersen. We reanalyzed the data and did more control experiments in WT mice. We found that light inactivation of S1 reduces whisking initiation, as shown in Figure 4 of Sreenivasan et al. (2016). These results are shown below (Fig. R4, panels b and d, this figure was added as Supplementary Fig. 7). This figure includes two more panels that also addressed a concern raised by the third reviewer (panels a and c).

Figure R4: The effect of light stimulation on whisking initiation in WT and GAD-ChR2 mice. **a.** Probability of whisking initiation in WT mice following observing no-whisking during the 0.5 seconds before light onset was compared to trials in which no light was delivered (control). **b.** Probability of whisking initiation following observing no-whisking in the 0.5 seconds before light onset for WT mice and GAD-ChR2 mice. **c.** Average whisking signal in WT mice for low whisking (purple) and high whisking (teal). Whisking trials were sorted similarly to the analysis of the data in Figure 3. **d.** The same as in C but for the GAD-ChR2 mice. Note that for both conditions and particularly for the low-whisking case (purple line), the whisking pattern was not changed during the first 200 milliseconds after light onset. The period at which we observed a transient hyperpolarization in cells located in the hemisphere contralateral to the light-inhibited side.

The probability of whisking initiation is shown in Figure R4 (Supplementary Fig. 7) for these two comparisons:

In panel **a** of Figure R4 we show that illuminating the barrel cortex of WT mice had no measurable effect on whisking initiation (the third reviewer raised the possibility that mice could see the light and initiated whisking). This was checked by selecting trials in which no whisking was detected in the first 0.5 seconds and then calculating the probability of observing whisking initiation in the following 1 s in control trials (without light) or during light stimulation. As can be observed, light did not change the probability of whisking initiation.

In panel **b** we compared the distribution of the latter measurements (i.e., probability of initiation of whisking following the light in WT mice) to that in GAD-ChR2 mice (i.e., similar to the study of Sreenivasan). Here we found, like in the results of Sreenivasan et al., that the probability of whisking initiation was reduced by light. These findings raised a concern that the transient hyperpolarization we observed when we illuminated the cortex during quiet periods (Fig. 4d in the original manuscript) could result from a change in whisking pattern. However, as shown in panel **d** of this Supplementary Figure, the light did not affect whisking in the first couple hundred milliseconds following light onset, where the clear transient hyperpolarization was observed in our intracellular recordings during low-whisking periods. This was done by averaging all whisking signals for all the trials (not only those showing no whisking in the first 0.5 seconds), based on low-whisking and high-whisking states, as done in Figure 4, which showed no clear change in mean whisking signal. The same is also true for the WT mice (panel c). Hence, despite the reduced probability of whisking initiation, we didn't find any evidence for a change in whisking pattern, in particular when inspecting the average whisking signal in the first 200 milliseconds from the light onset **in the low-whisking trials** (purple curve in **d**) – a period in which a transient hyperpolarization was observed. We summarized these findings in the Results section as follows:

“The effect of the light stimulation on the contralateral membrane potential could potentially result from a change in the whisking pattern, either as an arousing signal or due to blockade of whisking initiation signals from S1 to M1²⁸. Illuminating the cortex in WT mice showed that the probability of whisking initiation was not changed by light (Supplementary Fig. 7A), indicating that the arousal state was not affected by the light stimulation itself. However, in agreement with Sreenivasan et al. 2016, inactivation of the barrel cortex reduced the probability of whisking initiation in GAD-ChR2 mice when inspected in the first 1000 milliseconds following light onset (Supplementary Fig. 7b). Yet, we found no measurable effect of the light stimulation on whisking pattern in the first 200 milliseconds following light onset in the low-whisking trials, where a clear transient hyperpolarization was observed in the membrane potential activity (Fig. 3c-e). Hence, this transient hyperpolarization cannot be explained by a change in whisking pattern.”

- The authors assume that corpus callosum is connecting the entire area of the barrel cortices, is there any clear evidence for such mapping? In the visual cortex of rodents, cats and monkeys, corpus callosum connect only small parts of the visual cortex, parts that have their receptive fields along the vertical meridian. If corpus callosum also connects only specific parts of the somatosensory cortices in mice, the exact site of recordings/inactivation become much more important and can strongly influence the results.

Thanks for this question, as it helps to clarify this matter. In the somatosensory cortex, callosal projections are not only homotopic (as explained below in the section that was added to the Discussion). Yet, as described in the introduction, previous studies (for example from the lab of Timothy Murphy) showed prominent interhemispheric correlations between right and left primary somatosensory cortices. While it is possible that the correlations between the two somatosensory cortices are further constructed based on the connectivity between specific subareas in right and left barrel cortices, our anatomical data lacks the resolution to address this possibility. Yet our LFP recordings that were made across the two hemispheres show no differences in interhemispheric correlations between homotopic or non-homotopic barrels (Supplementary Fig. 2). This was written in the original submission (“Because we found that interhemispheric correlations were not related to the barrel maps, laminar probes were inserted without targeting specific barrel columns (Supplementary Fig. 2)”). Hence, we strongly suggest that interhemispheric correlations during non-whisking and whisking periods are weakly dependent on the specific locations within the right and left barrel cortices. Nevertheless, we added this paragraph into the Discussion, which deals with the pattern of callosal projections in this system:

“Moreover, it is known that in addition to callosal projections between homotopic barrel cortices, denser projections are found onto more lateral somatosensory areas, near the border between S1 and S2^{15,29,30}. This raises the possibility that our chemogenetic silencing would result in an even a greater effect if CNO was applied to more lateral areas, blocking any contralateral inputs to these areas which can propagate back to the recorded barrel cortex. In addition, it is possible that callosal projections between the two barrel cortices are specific based on the barrel maps. Yet, using paired LFP recordings from homotopic and non-homotopic barrel columns (Supplementary Fig. 2) we failed to find a fine structure in the interhemispheric correlations during ongoing and whisking periods.”

- In a related issue, roughly how much cortical volume was inhibited with light and CNO?

A recent paper by Doron et.al (2020, Larkum’s lab) shows that following the application of CNO, applied similarly as in our study, it spreads into deep layers of the cortex, although to a much lower degree compared with the upper layers. Hence, it is still possible that CNO did not fully silence all the callosal axons.

Here is a copy of that text and figure from Doron et al. 2020:

Our intracellular recordings during optogenetic inhibition (old Fig.4, now Fig. 3), as well as in our previous study¹⁹, show that most synaptic inputs to the recorded cells were diminished. In our previous study¹⁹, we demonstrated that L6 neurons are also inhibited by this light. We estimated that the volume of the cortex that was inhibited was as follows: The optical stimulation was given via a 400 μm fiber at a distance <2 mm from the cortex. According to that, the inhibited cortical area was approximately of a circle with a 1.54 mm diameter ($\sim 1.86 \text{ mm}^2$) centered at the barrel field (1.3 C, 3.3 L). Hence, most barrels inhibited were clearly homotopic to the site of recording on the other side. We refer to this information in the Methods section as follows:”

Optogenetic stimulation

For optogenetic stimulation, we used an analog modulated blue DPSS laser ($\lambda = 473 \text{ nm}$, Shanghai Dream Lasers Technology) coupled to a multimode fiber ($\text{\O}400 \mu\text{m}$ core, 0.39 NA, FT400UMT, ThorLabs, Newton, NJ). The intensity of the light was $\sim 7 \text{ mW}$ at the tip of the fiber. For local cortical stimulation, a continuous light pulse 2 seconds long was given through the fiber that was placed < 2 mm from the surface of the barrel cortex (1.3 C, 3.3 L from bregma). “

- Fig. 4b: the traces from both hemispheres before and after inactivation seem to be different from each other. Also, I was surprised for what seems like a complete lack for a post-inhibitory ‘rebound’ that typifies optogenetic-based inhibition.

We agree with the reviewer that there is some variability in the membrane potential activities of these two cells, but this kind of differences in Vm dynamics across cells is not new in the field and perhaps reflects the possibility that recordings were made from different types of neurons or at slightly different positions both in XY and depth coordinates.

With respect to the lack of post-inhibitory ‘rebound’, we know that it is prominent in slice preparations but might not be as prominent in awake mice. The lack of a rebound in our results is in line with other studies^{31–33}. We did not see a rebound effect in our experiments but rather a somewhat slow recovery from the light inactivation.

- Lines 479-484: is there any evidence for the authors suggestion on the relationship between corpus callosum modulation during quiet epochs and homeostasis?

These are only speculations, and most of them are in line with similar speculations raised in other published studies which we cited. In particular, studies from the labs of Tim Murphy and Rafi Malach raised similar hypotheses which we extended and proposed for the quiet wakefulness epochs. If the reviewer thinks that this is too speculative we can soften our hypothesis.

Minor:

- Fig 1: the letter 'b' is missing for the second panel.

fixed

- Fig.2: The cross- correlations are too small to infer where exactly the peak is and its relationship to zero time-lag.

We have added a supplementary figure 5, which contains the correlation of all recorded pairs with a magnified view of the correlation (column C), so it will be easy to infer the time of the peak in relation to zero.

Reviewer #3 (Remarks to the Author):

The study presented in the manuscript # NCOMMS-20-38247-T by Oran, Katz and collaborators, investigated the implication of transcallosal projections in interhemispheric correlation of neuronal activity and its modulation by the behavioral state, in awake head-fixed mice.

Focusing their attention on the whiskers representation within the primary somatosensory cortex (S1), they used complementary experimental approaches: single-unit, whole-cell and local field potential bilateral recordings, in addition to opto and pharmacogenetic approaches. From this substantial set of data, they demonstrate a decrease of interhemispheric correlation of neuronal activity at both supra and sub-threshold levels upon whisking. They further show results indicating that this decrease of interhemispheric correlation is likely to be due to a reduced recruitment of transcallosal projections in this behavioral state.

These two observations are novel and as such of interest for the community. Indeed, on the one hand, a strong synchronization of spontaneous waves of activity between hemispheres through transcallosal projections during the awake “quiet” state has been reported before by means of mesoscale imaging experiments (Mohajerani et al., 2010). On the other hand, a transition from “quiet” to “whisking” behavior has been shown to be associated with a disappearance of these slow waves, together with membrane potential depolarization and local neuronal desynchronization (Crochet & Petersen, 2006, Poulet & Petersen, 2008, Poulet et al., 2012; Eggermann et al., 2014). Given such local desynchronization and depolarization during whisking, which has been shown to emerge from an increase of both thalamic and cholinergic drive, one could have expected that measuring long-range interhemispheric correlation would be made more

difficult, but not necessarily that whisking would be associated with a decrease of activity in transcallosal axons.

The study reported here therefore provide data of interest and is globally technically sound, however, some control experiments are needed to support the main conclusions and the quality of the manuscript has to be improved. Here a list of concerns that deserved to be addressed.

- Mice are usually, after a few sessions of habituation to head fixation, rarely whisking spontaneously, and could even present episodes of sleep (Fernandez et al., 2017). In the present study, however, the authors do not mention any habituation of the animals to the head fixation before the recordings. This might explain why they assume that the mice are never falling asleep during the recordings, splitting the data in only two states: quiet awake and whisking. This might also be why they could get relatively frequent episodes of whisking during the recordings. However, the proportion of time spend in quiet vs whisking behavior should be clearly stated for all the figures. In addition, in case any special experimental means have been used to trigger whisking episodes during the recordings, this should be clearly mentioned in the methods section.

This is indeed a relevant question. Therefore, we added analysis to fully address this issue. Mice were habituated for 30-60 minutes before starting the recording or imaging sessions. Furthermore, in all the experiments except for the extracellular recording, mice were able to freely walk on a treadmill. In a subset of experiments, we used a camera to track the pupils. When examining these movies, we did not reveal any tendency for the mice to fall asleep (eyes closed) during the recording sessions. As requested, we analyzed the proportion of time the animal spent in quiet vs whisking epochs for every animal and found that the animal spends more time in quiet epochs compared to whisking (added as Supplementary Figure 3, also shown below as Figure R5). In the figure where we describe the results of this analysis short transition times were excluded, hence the sum of whisking + quiet is slightly less than 100%. These data show that on average mice spent about 14%-30% of the time whisking. We also never observed prolonged quiet (non-whisking) periods that suggest that mice were asleep (across all our experiments the longest quiet period was 24 s). Our data therefore indicate that whisking epochs were less frequent and shorter in duration than quiet epochs. Lastly, no special experimental means were used to trigger whisking epochs.

Figure R5 Percentage of time mice spent in quiet vs. whisking states at each experiment shown in the main Figures of the manuscript. The proportion of time spent in whisking state (teal bars) varied from 14% (Figure 3) to 30% (Figure 4).

- Given this experimental design, it is likely that episodes of whisking are correlated with global body movements of the mouse trying to escape from this unusual head fixation condition. To my point of view, the expression “exploratory” is not really appropriate to qualify this behavior.

Indeed, similar ongoing studies in our lab, in which locomotion is monitored in head-fixed mice, have shown that whisking is correlated with global body movements. Since locomotion was monitored only in a small number of the recording sessions of the current study, we have not included this data.

As exploration is an activity mouse naturally does while freely moving, we agree that the expression “exploratory” is not appropriate and we replaced it with “active behavior”.

- The figure 6 and movie 1 illustrate 2-photon calcium imaging of transcallosal axons in awake mice. These data are supporting the main conclusion of the manuscript: a decrease of transcallosal activity during whisking. As acknowledged by the authors, and for the reasons mentioned above, the whisking episodes reported here are likely to be accompanied by strong movements that might impact the 2-photon recording of calcium transients in thin axonal portions. Performing similar experiments, with a viral construct driving the expression of GFP instead of GCamp6s, would allow assessing how these movements affect the axonal imaging during whisking and if they could ultimately lead to a misinterpretation of the results.

This is an excellent question and as requested we performed control experiments in which we expressed GFP in callosal axons and measured the signals from 15 axons. Analysis of these new data shows that GFP axonal signals were very stable and showed no correlation with whisking movements (Fig. R6 and as a new Supplementary Figure 12). The following text was added to the results:

“Control imaging experiments in awake mice expressing GFP in callosal axons (Supplementary Fig. 12) showed no correlation between GFP signals and whisking. This suggests that the negative correlation observed between calcium signal and whisking signal was not caused by movement artifacts.”

Figure R6: **a.** Schematic diagram of the injection and imaging procedures. AAV-GFP was injected in the left barrel cortex to express GFP in CaMKII cells and imaging of the contralateral hemisphere was performed three weeks later. **b.** Example of simultaneous imaging of an axon (green), neuropil (gray) together with the whisking signal (teal). **c.** Cross-correlation between the callosal axonal GFP signals and whisking (green) and between the callosal axonal GCaMP6s and whisking signals (brown). The latter is the same correlation that is presented in Figure 5c. **d.** Mean fluorescence GFP axonal signals for quiet and whisking epochs ($n = 15$). **e.** Histogram of the modulation index (i.e., effect of whisking on the GFP signal) for the data presented in d.

- The methods section indicate that pupil diameter has been monitored together with whisking activity during these imaging experiments (line 674), and that the mice were held on a treadmill (lines 672 & 676). Either these mentions have been made by mistake (text inherited from a previous unrelated work), or, if it is really the case, they deserve to be more clearly stated and commented in the main text and complementary analyses should be provided. From what I have understood, there was no treadmill for all the other experiments of the study, which means that the behavioral “whisking” state is likely to be different in this last set of experiments, being most probably associated with running.

We thank the reviewer for raising this issue. A treadmill was indeed used in all the experiments, except for the extracellular recordings, but locomotion speed was not monitored and therefore was not reported.

We monitored the pupil only in some of the imaging experiments and although initially we thought to include this analysis (explaining why this mention was left in the text), due to technical problems, we decided eventually not to use this monitoring for the analysis. In all the other types of experiments, we did not monitor the pupil size, and therefore we decided to remove

any text in which pupil monitoring was mentioned. Thank you for drawing our attention to this issue.

- In addition, there are many details in this figure that should be clarified. Lines 685-687 of the methods section, it is indicated that a different preprocessing has been applied to the “illustrated examples” than “in all other analyses”, why?

In the illustrated examples we presented signals that were normalized as $\Delta F/F$ and in the population analysis the signals were processed as $\text{norm}F$ (normalization of all the signals between 0 to 1) in order not to over- or under-weight highly active calcium traces, similar to what has been done in a previous study⁴. However, following the reviewer note, we examined how the normalization of $\Delta F/F$ affects the results and found it had no quantitative effect on the results, except for the magnitude (y-axis) of the signals in the two types of the analyses. The cross-correlation was not affected as it is already a normalized analysis, showing correlation coefficients. However, in the second analysis that is dependent on a threshold to capture high levels of axonal activity (as described in the results line 331), the use of $\Delta F/F$ for signals with low/high variability will give over- or under-weighted results. $\text{Norm}F$ allows maintaining all the signals between 0 to 1 and therefore equalizes all different axons signals. Below we present in the figure the differences between the two types of normalization (Fig. R7):

Figure R7. The two quantification calculations of the axonal calcium signals provide very similar results. a. $\Delta F/F$ method when measuring the mean signals above a threshold. b. $\text{Norm}F$ method when measuring the mean signals above the threshold. Note that statistical tests gave almost the same P values (rank-sum).

The two examples illustrated in panel e have been recorded simultaneously. Why not illustrating the activity of the two axonal portions throughout the same episodes of whisking then? Given the large variability of the population data, illustrating examples from different mice would be very welcome.

From the above comment, we understand that the legend in this figure was misleading. The two examples in panel C were imaged from two different animals; we clarified this in the Figure 5 legend (Figure 6 in the original submission).

- The panel a shows a strange assembly of images, are they all coming from the same histological section? Please specify.

These three images were taken from the same slice but with different light exposure levels. This was done to expose the axons which exhibited a lower fluorescent signal when compared to the signal level at the injected area. We framed each image and clarified it in the figure legend.

- The movie is lacking any temporal or spatial scale indication.

The movie was updated and now it includes a scalebar and time scale for the abscissa.

- The results in figure 4 are interesting however control experiments assessing the effect of light in the absence of ChR2 expression would be very useful.
- It is also crucial to provide a proper analysis of the behavior to check that the photostimulation is not triggering any whisking when delivered in the quiet state. The extended fig. 4 in its actual form is non informative, please specify what are the plots corresponding to, and provide raw data measurement of whisking activity during the photostimulation.

We want to thank the reviewer for this important suggestion. We performed control experiments for testing whether the light within the craniotomy (the same protocol as used with the GAD-ChR2 mice) over the barrel cortex initiates whisking during quiet trials (Fig. R8, $n = 8$ sessions, in 3 WT animals). Trials were sorted by the whisking activity before light onset ($-0.5s - 0s$, also in the control trials where the light was off), and the lower quintile trials ($n = 16$ out of 80) were used for analysis (Fig. R8, raw data as requested). The STD of the whisking signal in control trials without light (Fig. R9, green bars) and during trials with light ($0s - 0.5s$, purple bars) were compared. No difference in whisking was found during the light period at any single experiment. Summary of these results show that the whisking probability in quiet trials with and without light stimulation was similar (Fig. R9 - lower right panel, $p > 0.7$, WSRT). The statistics of these experiments are also shown in Figure R4 (a and c, and presented as additional Supplementary Figure 7) as part of an answer to a second reviewer's comment.

We also evaluated if light affects whisking in general, by comparing the whisking before and following light initiation (in all trials), and found no difference (Fig. R10).

Figure R8 – Example traces of whisking activity from 8 experiments performed in WT mice, showing the quintile of trials with the lowest activity before light (20%, $n = 16$ trials, blue traces) and in control trials (lowest activity, no-light, black traces) when no light was given after a 0.5 s period of no whisking. The bottom right panel shows the normalized (Z score) population average traces, indicating no difference for the average whisking signals when comparing data obtained with light (blue trace) and without light (black trace).

Figure R9: The effect of optogenetic light on whisking initiation in WT mice in quiet trials. The first 8 panels show the amount of whisking after light onset (purple bars, 8 sessions in 3 mice corresponding to the raw data in Fig. R7) when considering only the lower quintile of trials according to the amount of whisking during 0.5 seconds prior to light onset compared to control trials in which no light was delivered (green bars). The bottom right panel shows the probability of whisking initiation for these experiments.

Figure R10: Whisking activity before and during the light periods was similar. In all trials with optogenetic stimulation ($n = 80$), whisking activity before light ($-0.5 \text{ s} - 0 \text{ s}$, green bars) was not different from that during light ($0 \text{ s} - 0.5 \text{ s}$, purple bars, WSRT).

- Is the quantification in panel d corresponding to a mean V_m in the first 200ms following light or a negative peak V_m value?

The quantification in panel d corresponds to the mean change in V_m , 200ms following light relative to the baseline. This is mentioned in the manuscript (“ ΔV_m = Difference between mean V_m during first 200 ms following light inactivation and mean baseline V_m before light (800ms), mean \pm SEM, $p < 0.05$, WSRT”).

- Which time window has been used for “baseline” measurement?

To evaluate the baseline membrane voltage we calculated the mean V_m for the 800 ms before light. We’ve added this value to the updated Figure 3 legend.

- Why not showing the whisking activity in panel b?

In the updated Figure 3 (Figure 4 in the original submission), whisking signals were added below the membrane voltage traces as suggested.

- In figure 5, panel b and d, it should be specified if these are example STAs for given units or population averages. If examples, then it should be indicated to which line they correspond in the next panel.

Figures 5b, d (now Figures 4b, d) are population STA, while c and e show the STA of all pairs during quiet wakefulness (c) and whisking (e). The figure legend was changed to indicate that panels b and d refer to population effect.

- One could expect that the CNO application should not affect the “within hemisphere” STAs as opposed to the inter-hemisphere ones. It would be nevertheless interesting to show it.

Figure R11: The effect of CNO application on local correlation. STAs were calculated before (color lines) and after the application of CNO (gray lines). **a-b** Local STAs for whisking and quiet epochs before application of CNO as recorded in the hemisphere in which CNO was applied (right hemisphere). **c-d** Local STAs for whisking and quiet epochs before applying CNO as recorded in the left hemisphere, where hM4D was expressed (Supplementary Figure 10).

This is an important question that led us to reanalyze these recordings. As the reviewer suggested, we also expected to find no change in the ‘within’ hemisphere STAs. Therefore, we were surprised to find that the STAs within hemispheres were altered, although to a lesser degree when compared to the interhemispheric STAs. Actually, and in contrast to our intuition, the major change we found was a small reduction in the STA during the non-whisking epochs in the hemisphere in which the virus was injected. The changes in the STAs within the hemispheres are presented as Figure R11 and also as Supplementary Figure 10.

Figure R12. STAs across and within hemispheres were stable during recording sessions. A-D Four combinations of superpositions of STAs taken from first and second portions of the data, both before (green) and after (red) application of CNO to the “within CNO hemi” (right barrel cortex). Virus was injected in the “non-CNO hemi” (left hemisphere). The portions of the data for each of the 4, is marked above each.

While we have no explanation for this observed change, we decided to carefully check the data and see if it was consistent across the entire recording session (i.e., to see if it was stationary or not). Changes in STA magnitude, within and across hemispheres, could reflect also a change in brain state during the entire session lasting before and after application of the CNO.

Thus, the first test was to split the data recorded before into two roughly equal segments, and do the same for the data recorded after CNO application. We used these partitions to compare all the 4 combinations. For example, we compared the 2st half before application of CNO to the 1st half after applying CNO (the closest portions in time). A similar check was made for the 2nd half before and 2nd after applying CNO and so forth. All these 4 comparisons show very similar effects (Figure R11), even when comparing non-overlapping portions of the data (i.e., panels a and d). Hence, we

conclude that the changes we observed following CNO application represent a stationary change rather than a contribution from specific portions of the data that were recorded either before or after the application of CNO. Note that in all the 4 combinations, we observed a marked reduction in interhemispheric STAs and some reduction in the ‘within hemisphere’ STA for the hemisphere in which hM4D was expressed. Note that a small elevation in within STA was also observed during whisking in the hemisphere in which CNO was added. However, changes were much more prominent across the two hemispheres during non-whisking epochs.

Figure R13: Whisking pattern was not affected by the application of CNO a-d. We compared the percentage of whisking time (a), mean duration of whisking epochs (b) standard deviation (c) and whisking amplitude (d) and in 7 mice. None of these properties were significantly different (rank-sum test). (Supplementary Figure 11).

In the second test we asked if the behavior of the mice was changed following the application of CNO, as such a change might in turn lead to the changes in the STAs. Therefore, we analyzed the whisking pattern of 7 mice, and asked if the relative whisking time, mean duration of whisking epoch (and its STD) and amplitude were changed following application of CNO (Figure R12). We found however that none of these parameters changed after application of CNO, strongly suggesting that the changes that we observed in the STAs were not caused by a shift in whisking behavior, a known proxy to brain state. Taken together, we found that 1) CNO had no effect on mean firing rate (Figure R1, response to the first reviewer), 2) the effect of CNO on STAs was consistent when partitioning the data, 3) whisking pattern was not altered. We thus strongly suggest that the prominent effect of CNO on interhemispheric correlations and its moderate effect on local correlations were caused by a direct effect of the CNO on callosal terminals rather than by other factors. We added to the text the following two sentences:

In Results:

p. 14: “In addition to the prominent reduction in the STA across hemispheres during quiet epochs, we observed a moderate reduction in that state also in the local STA measured from the activity in the left hemisphere, where hM4D was expressed (Supplementary Fig. 10). The reduction in local correlations perhaps reflects a loss of feedback to this hemisphere. Importantly, changes in interhemispheric and local correlations due to the application of CNO were not accompanied by any prominent changes in the whisking pattern (Supplementary Fig. 11).”

In the discussion:

p. 22: “Surprisingly, the application of CNO also led to a reduction in local correlations in the hemisphere in which it was injected. The underlying mechanisms for this change are not clear and might reflect a loss of feedback inputs from the hemisphere in which the callosal fibers were inactivated.”

- Minor points:

The text of the manuscript, and legends in particular must be carefully reread.

The figure 2 and 3 should be merged together.

Figures 2 and 3 were merged.

The authors could refer to an interesting recent related study: Ferrier J, Tiran E, Deffieux T, Tanter M, Lenkei Z. Functional imaging evidence for task-induced deactivation and disconnection of a major default mode network hub in the mouse brain. *Proc Natl Acad Sci U S A.* 2020 Jun 30;117(26):15270-15280.

Thanks for mentioning this recent paper. We now cite it in the introduction.

Some key aspects of the experiments are missing in the methods section:

- LFP recordings: how are the LFP bundles inserted in the craniotomies, at which depth?

We further explained the method and added the depth of recordings:

“Custom-made bundles for LFP recordings were built using formvar-insulated nichrome wire (#761500, AM systems, WA, USA). The recording bundles were coupled to an Intan RHD2132 headstage and data was sampled at 30 kHz using Open Ephys³⁴. Each bundle had a reference

channel and two signal channels inserted 550 μm beneath the dura using a micromanipulator (MX7600; Siskiyou, Grants Pass, OR) into miniature craniotomies over the A1 and D1 barrels. Data was further band-passed to 0.1-200 Hz and analyzed using custom MATLAB code (Mathworks, MA, USA). “

- Whisking detection via camera: the threshold algorithm seems to apply to a 1D “signal”, how is this signal related to the acquired 2D images?

Whisking detection via the camera was updated:

“To extract the whisking behavior from the video, an ROI in the central part of the whisker pad was selected using a custom LabVIEW graphical user interface which computed the squared difference in intensity between the pixels of each pair of consecutive frames. The whisking trace was defined as the sum of the squared differences for all pixels divided by the number of pixels within the ROI, i.e. the mean squared error between frames. The signal was smoothed with a Gaussian window ($\sigma = 200$ ms) and whisking episodes were detected using a threshold (1.5 times the upper 15th percentile of the whisking signal). Whisking episodes were identified as high amplitude changes in the signal crossing the threshold, with at least 500 ms in duration. Minimum times between two episodes were set to at least 500 ms. Episodes crossing the threshold but not meeting the duration and separation requirements were ignored. The rest of the data was defined as quiet wakefulness with a 100 ms buffer time between whisking and quiet wakefulness.”

- The lesion experiments should be described with more detail in a dedicated section.

We added a section in the methods about the lesion procedure.

“Animals previously mounted with headbars were anesthetized using isoflurane (1-2%), a 0.5mm x 0.5mm craniotomy between the two barrel cortices (on the midline, 2.4 mm caudal) was made. A glass pipette with a broken tip connected to a narrow suctioning pipe was inserted 1.4 mm deep using a motorized manipulator (MX7600; Siskiyou, Grants Pass, OR). After 5 minutes that the pipette was in place (to allow the brain to return to normal position), the baseline LFP was recorded for about 20 minutes. Following this recording, suction was turned on for a couple of seconds. 5 minutes following the end of suctioning the LFP was again recorded for 20 minutes.”

- The extended data file has to be corrected. The extended fig. 3 is not supporting the assertion of the main text (lines 149-151) where it is mentioned. The extended fig. 4 is missing a proper title and legend etc...

The extended data file has been corrected. We thank the reviewer and changed the reference to the Supplementary Fig. 3 in the correct place.

1. Urbain, N. *et al.* Whisking-Related Changes in Neuronal Firing and Membrane Potential Dynamics in the Somatosensory Thalamus of Awake Mice. *Cell Rep.* **13**, 647–656 (2015).
2. Yu, J., Gutnisky, D. A., Hires, S. A. & Svoboda, K. Layer 4 fast-spiking interneurons filter thalamocortical signals during active somatosensation. *Nat. Neurosci.* **19**, 1647–1657 (2016).
3. Deitcher, Y., Leibner, Y., Kutzkel, S., Zylbermann, N. & London, M. Nonlinear relationship between multimodal adrenergic responses and local dendritic activity in primary sensory cortices. *bioRxiv* 814657 (2019) doi:10.1101/814657.
4. Reimer, J. *et al.* Pupil fluctuations track rapid changes in adrenergic and cholinergic activity in cortex. *Nat. Commun.* **7**, 1–7 (2016).
5. Constantinople, C. M. & Bruno, R. M. Effects and mechanisms of wakefulness on local cortical networks. *Neuron* **69**, 1061–1068 (2011).
6. Durán, E., Yang, M., Neves, R., Logothetis, N. K. & Eschenko, O. Modulation of Prefrontal Cortex Slow Oscillations by Phasic Activation of the Locus Coeruleus. *Neuroscience* **453**, 268–279 (2021).
7. Goard, M. & Dan, Y. Basal forebrain activation enhances cortical coding of natural scenes. *Nat. Neurosci.* **12**, 1444–1449 (2009).
8. Pinto, L. *et al.* Fast modulation of visual perception by basal forebrain cholinergic neurons. *Nat. Neurosci.* **16**, 1857–1863 (2013).

9. Kinnischtzke, A. K., Fanselow, E. E. & Simons, D. J. Target-specific M1 inputs to infragranular S1 pyramidal neurons. *J. Neurophysiol.* **116**, 1261–1274 (2016).
10. Kinnischtzke, A. K., Simons, D. J. & Fanselow, E. E. Motor cortex broadly engages excitatory and inhibitory neurons in somatosensory barrel cortex. *Cereb. Cortex N. Y. N 1991* **24**, 2237–2248 (2014).
11. Yu, J., Hu, H., Agmon, A. & Svoboda, K. Recruitment of GABAergic Interneurons in the Barrel Cortex during Active Tactile Behavior. *Neuron* **104**, 412-427.e4 (2019).
12. Le Bé, J.-V., Silberberg, G., Wang, Y. & Markram, H. Morphological, electrophysiological, and synaptic properties of corticocallosal pyramidal cells in the neonatal rat neocortex. *Cereb. Cortex N. Y. N 1991* **17**, 2204–2213 (2007).
13. Lee, A. T. *et al.* Pyramidal neurons in prefrontal cortex receive subtype-specific forms of excitation and inhibition. *Neuron* **81**, 61–68 (2014).
14. Slater, B. J. & Isaacson, J. S. Interhemispheric Callosal Projections Sharpen Frequency Tuning and Enforce Response Fidelity in Primary Auditory Cortex. *eNeuro* **7**, (2020).
15. Petreanu, L., Huber, D., Sobczyk, A. & Svoboda, K. Channelrhodopsin-2-assisted circuit mapping of long-range callosal projections. *Nat. Neurosci.* **10**, 663–668 (2007).
16. Tagawa, Y. & Hirano, T. Activity-Dependent Callosal Axon Projections in Neonatal Mouse Cerebral Cortex. *Neural Plast.* **2012**, (2012).
17. Anastasiades, P. G., Marlin, J. J. & Carter, A. G. Cell-Type Specificity of Callosally Evoked Excitation and Feedforward Inhibition in the Prefrontal Cortex. *Cell Rep.* **22**, 679–692 (2018).
18. Palmer, L. M. *et al.* The cellular basis of GABA(B)-mediated interhemispheric inhibition. *Science* **335**, 989–993 (2012).

19. Cohen-Kashi Malina, K., Mohar, B., Rappaport, A. N. & Lampl, I. Local and thalamic origins of correlated ongoing and sensory-evoked cortical activities. *Nat. Commun.* **7**, 12740 (2016).
20. Okun, M., Naim, A. & Lampl, I. The subthreshold relation between cortical local field potential and neuronal firing unveiled by intracellular recordings in awake rats. *J. Neurosci. Off. J. Soc. Neurosci.* **30**, 4440–4448 (2010).
21. Guevara, M. A. & Corsi-Cabrera, M. EEG coherence or EEG correlation? *Int. J. Psychophysiol.* **23**, 145–153 (1996).
22. Reig, R. & Silberberg, G. Distinct Corticostriatal and Intracortical Pathways Mediate Bilateral Sensory Responses in the Striatum. *Cereb. Cortex N. Y. N 1991* (2016) doi:10.1093/cercor/bhw268.
23. Curtis, J. C. & Kleinfeld, D. Phase-to-rate transformations encode touch in cortical neurons of a scanning sensorimotor system. *Nat. Neurosci.* **12**, 492–501 (2009).
24. Kock, C. P. J. de & Sakmann, B. Spiking in primary somatosensory cortex during natural whisking in awake head-restrained rats is cell-type specific. *Proc. Natl. Acad. Sci.* **106**, 16446–16450 (2009).
25. Fanselow, E. E. & Nicolelis, M. A. Behavioral modulation of tactile responses in the rat somatosensory system. *J. Neurosci. Off. J. Soc. Neurosci.* **19**, 7603–7616 (1999).
26. Ferezou, I. *et al.* Spatiotemporal Dynamics of Cortical Sensorimotor Integration in Behaving Mice. *Neuron* **56**, 907–923 (2007).
27. Gentet, L. J., Avermann, M., Matyas, F., Staiger, J. F. & Petersen, C. C. H. Membrane potential dynamics of GABAergic neurons in the barrel cortex of behaving mice. *Neuron* **65**, 422–435 (2010).

28. Sreenivasan, V. *et al.* Movement Initiation Signals in Mouse Whisker Motor Cortex. *Neuron* **92**, 1368–1382 (2016).
29. Wise, S. P. & Jones, E. G. Developmental studies of thalamocortical and commissural connections in the rat somatic sensory cortex. *J. Comp. Neurol.* **178**, 187–208 (1978).
30. Suárez, R. *et al.* Balanced interhemispheric cortical activity is required for correct targeting of the corpus callosum. *Neuron* **82**, 1289–1298 (2014).
31. Ausborn, J. *et al.* Organization of the core respiratory network: Insights from optogenetic and modeling studies. *PLoS Comput. Biol.* **14**, (2018).
32. Cohen-Kashi Malina, K., Mohar, B., Rappaport, A. N. & Lampl, I. Local and thalamic origins of correlated ongoing and sensory-evoked cortical activities. *Nat. Commun.* **7**, 12740 (2016).
33. Ledri, M., Madsen, M. G., Nikitidou, L., Kirik, D. & Kokaia, M. Global Optogenetic Activation of Inhibitory Interneurons during Epileptiform Activity. *J. Neurosci.* **34**, 3364–3377 (2014).
34. Siegle, J. H. *et al.* Open Ephys: an open-source, plugin-based platform for multichannel electrophysiology. *J. Neural Eng.* **14**, 045003 (2017).

Reviewer #1 (Remarks to the Author):

The Authors have thoroughly revised the manuscript and addressed all of my comments. I wish to congratulate them for an interesting and timely study.

Reviewer #2 (Remarks to the Author):

The authors have addressed all my concerns.

Reviewer #3 (Remarks to the Author):

Overall the authors have made a valuable effort to address the points raised on the first version of the manuscript. The article has consequently improved a lot in quality.

In particular, this is a great add to the paper to have performed control experiments with GFP instead of GCamp6s to assess how movements (most probably often associated with whisking), affect the axonal imaging. However, lines 564-573 of the main text, the animal(s) used for this experiment should be indicated as well as the viral construct (also lines 589-590). On the Sup. Fig. 12, showing an image of a field of view taken from whisker S1 contralateral to the injection site (as in fig. 5 for the GCamp6s) would be very welcome to strengthen these results. In the main text has been added the sentence: "Control imaging experiments in awake *mice* expressing GFP in callosal axons". However, I am wondering if the 15 axons imaged are coming from one mouse, one field of view (FOV), same recording, one mouse, several FOV and recordings, or more than one mouse? Please specify the number of animals & FOV used to construct this figure.

Few other minor points to be addressed:

- Figure 3 b: whisking signals were added, so the legend should be updated accordingly.
- Figure 5 b: in the legend, the text in brackets must be relocated just after the first sentence "Histology of the injection (left) and callosal projecting axons (right)" since it has nothing to do with the 2photon images on the two right most panels.
- Sup. Fig. 2: In the previous version of the manuscript, the intrinsic imaging response locations were superimposed on the images of the blood vessel patterns taken under green light. These have disappeared in the revised manuscript without any obvious reason. I think they should come back, even though the A1/D1 responses could look a bit closer than expected, the data are as they are and should not be hidden.... Actually those were justifying the colorcode of the pipettes on the panel c, which could be further used to help the reader dissociating in the data below the intrahemispheric correlations (blue or red) vs interhemispheric correlations (purple...).
- Sup. Fig. 3: On this figure (or at least in its legend), it would be very informative to indicate which figure corresponds to experiments where the mice were free to run on a treadmill (if I've understood well, fig 2,3, and 5) vs experiments where the mice were placed on a fixed support (fig 1 and 4).
- Sup. Fig. 10: panel labels should appear in bold (both fig and legend). "(Supplementary Figure 10)." Should be removed at the end of the legend.
- Sup. Fig. 11: In the legend there is an extra "and" to be removed.
- In the methods section for the lesion procedure, the authors now specify (line 624) that the micropipette was inserted at "2.4mm caudal", which seems very posterior regarding to S1Bf and to the histological slice shown in Sup. Fig. 8 (~1 mm caudal). Is it really the right coordinate? Was the

pipetted inserted with a given angle? Did the authors base their method on a published protocol or the mouse connectivity atlas? It would be welcome to mention a reference here or justify this location in one way or another...

- Sup. Fig. 13 a is missing a proper scalebar. Are the neurons in b coming from the ROIs illustrated in a? If yes, why not showing the ROIs in a with colors corresponding to the traces in b?

Response to the Third reviewer

We deeply appreciate the important comments of the reviewer that we addressed as shown below.

**Minor changes of figures were made on Supplementary figures 2,12,13.*

Reviewer #3 (Remarks to the Author):

Overall, the authors have made a valuable effort to address the points raised on the first version of the manuscript. The article has consequently improved a lot in quality. In particular, this is a great add to the paper to have performed control experiments with GFP instead of GCaMP6s to assess how movements (most probably often associated with whisking), affect the axonal imaging.

However, lines 564-573 of the main text, the animal(s) used for this experiment should be indicated as well as the viral construct (also lines 589-590).

We thank the reviewer for raising this issue, we added this information in the 'Methods\Animals' section:

“Control imaging of GFP expression in callosal axons was performed in two C57BL/6 animals injected with AAV-CaMKIIa-EGFP (Addgene, #50469-AAV5).”

And along with the injected constructs at the 'Viral injections' section:

“virus (AAV2/5-hSyn-hM4D(Gi)-mCherry, AAV-Syn-GCaMP6s-WPRE-SV40 or AAV-CaMKIIa-EGFP),”

On the Sup. Fig. 12, showing an image of a field of view taken from whisker S1 contralateral to the injection site (as in fig. 5 for the GCaMP6s) would be very welcome to strengthen these results.

Thanks for suggesting this addition. We have updated Supplementary figure 12 and added an example image of the callosal axons.

In the main text has been added the sentence: “Control imaging experiments in awake *mice* expressing GFP in callosal axons”. However, I am wondering if the 15 axons imaged are coming from one mouse, one field of view (FOV), same recording, one mouse, several FOV and recordings, or more than one mouse? Please specify the number of animals & FOV used to construct this figure.

We added this information at the legend of supplementary figure 12d.

“**d.** Mean fluorescence GFP axonal signals for quiet and whisking epochs (Two animals, n = 15 axons).”

Few other minor points:

- Figure 3 b: whisking signals were added, so the legend should be updated accordingly.

The legend was updated:

“**b.** Example traces for paired intracellular recordings from right (orange) and left (blue) hemispheres shown together with whisking trace (teal).”

- Figure 5 b: in the legend, the text in brackets must be relocated just after the first sentence “Histology of the injection (left) and callosal projecting axons (right)” since it has nothing to do with the 2photon images on the two right most panels.

We changed the legend to match the panels:

“ **b.** Histology of the injection (leftmost) and callosal projecting axons (second and third from the left), images were taken from the same slice with different exposure levels. The two right most panels show an active and non-active GCaMP6s-expressing axon.”

- Sup. Fig. 2: In the previous version of the manuscript, the intrinsic imaging response locations were superimposed on the images of the blood vessel patterns taken under green light. These have disappeared in the revised manuscript without any obvious reason. I think they should come back,

even though the A1/D1 responses could look a bit closer than expected, the data are as they are and should not be hidden.... Actually those were justifying the colorcode of the pipettes on the panel c, which could be further used to help the reader dissociating in the data below the intrahemispheric correlations (blue or red) vs interhemispheric correlations (purple...).

We thank the reviewer for notifying this error. We did not change the figure. The absence of the superimposed locations of the barrels resulted from file conversions when we've inserted the EPS image into the PDF file. The error was fixed.

- Sup. Fig. 3: On this figure (or at least in its legend), it would be very informative to indicate which figure corresponds to experiments where the mice were free to run on a treadmill (if I've understood well, fig 2,3, and 5) vs experiments where the mice were placed on a fixed support (fig 1 and 4).

The experiments presented in figure 2,3,5 were made at the same experimental setup; in head-fixed mice free to run on a treadmill, while experiments in figure 1 and 4 were made with fixed support. We added the text below in the manuscript to clarify that:

“To test the strength of coupling between the two hemispheres and to find how it depends on the behavioral state, head-fixed GAD-ChR2 transgenic mice free to run on a treadmill were used to optogenetically inhibit the activity of the barrel cortex in one hemisphere while we intracellularly recorded from the other homotopic area.”

And to supplementary figure 3 legend:

"In the experiments presented in figures 2,3 and 5, the animals were head-fixed and free to run on a treadmill, while the head-fixed animals of the experiments presented in figures 1 and 4 were on a fixed support.”

And to Figure's 3a legend:

“**a.** Paired intracellular recordings in the right and left barrel cortices of head-fixed GAD-ChR2 mouse on a linear treadmill.”

- Sup. Fig. 10: panel labels should appear in bold (both fig and legend). “(Supplementary Figure 10).” Should be removed at the end of the legend.

Done.

- Sup. Fig. 11: In the legend there is an extra “and” to be removed.

Removed.

- In the methods section for the lesion procedure, the authors now specify (line 624) that the micropipette was inserted at “2.4mm caudal”, which seems very posterior regarding to S1Bf and to the histological slice shown in Sup. Fig. 8 (~1 mm caudal). Is it really the right coordinate? Was the pipette inserted with a given angle? Did the authors base their method on a published protocol

or the mouse connectivity atlas? It would be welcome to mention a reference here or justify this location in one way or another...

We thank the reviewer for notifying this mistake. Following the lesion of the corpus callosum connecting the barrel cortices (1.3mm caudal to bregma), we observed a drastic decrease in spontaneous activity. To test if it results from a global decrease in brain activity, we also made control lesions at 2.4mm caudal to bregma (corresponding to callosal visual fibers). We mistakenly wrote this position. In both cases we found decreased activity at the barrel cortices, pointing toward a global decrease in firing due to callosal lesions. Since the callosal-barrel lesion experiments were already off the course of our question we didn't refer to the lesion experiments in the visual cortex.

- Sup. Fig. 13 a is missing a proper scalebar. Are the neurons in b coming from the ROIs illustrated in a? If yes, why not showing the ROIs in a with colors corresponding to the traces in b?

A scalebar and colors corresponding to the traces in b were added in panel a. The legend was updated accordingly.

a. A picture of the imaged cortex created using max projection in Suite2P (0.9.3, Pachitariu et al., 2017). **b.** Representative calcium signals from 8 cells and the whisking signal below (teal). **c.** The population average of the cross-correlations between whisking and calcium signals (n = 2 mice, n = 229 cells). **d.** Mean norm calcium signals that crossed a threshold (4 times the median absolute deviation) for quiet and whisking epochs (n = 229 cells). Scalebar = 200 micrometers.

Reviewer #3 (Remarks to the Author):

All the points listed before have been addressed properly.

Note that a scale bar is missing for the image you added in supfig 12 a. Furthermore, you've added the caption "Two animals, n=15 axons" but it would have been more informative to indicate how many axons per mice there were.